# Modularity of the segmentation clock and morphogenesis

**James E Hammond[1], Ruth E Baker[2], Berta Verd[1]***

[1]Biology Department, University of Oxford, Oxford, United Kingdom; [2]Mathematical Institute, University of Oxford, Oxford, United Kingdom

## eLife Assessment

This **valuable** manuscript uses mathematical modeling to address the synchrony of the vertebrate segmentation clock with the developmental processes. The authors use **convincing** arguments to support the idea that this would allow the evolution of flexible body plans and a variable number of segments. This manuscript could be of interest to developmental biologists and systems biologists. [Editors' note: this paper was reviewed by Review Commons.]

**Abstract** Vertebrates have evolved great diversity in the number of segments dividing the trunk body, however, the developmental origin of the evolvability of this trait is poorly understood. The number of segments is thought to be determined in embryogenesis as a product of morphogenesis of the pre-somitic mesoderm (PSM) and the periodicity of a molecular oscillator active within the PSM known as the segmentation clock. Here, we explore whether the clock and PSM morphogenesis exhibit developmental modularity, as independent evolution of these two processes may explain the high evolvability of segment number. Using a computational model of the clock and PSM parameterised for zebrafish, we find that the clock is broadly robust to variation in morphogenetic processes such as cell ingression, motility, compaction, and cell division. We show that this robustness is in part determined by the length of the PSM and the strength of phase coupling in the clock. As previous studies report no changes to morphogenesis upon perturbing the clock, we suggest that the clock and morphogenesis of the PSM exhibit developmental modularity.

***For correspondence:**
berta.verdfernandez@biology.ox.ac.uk

**Competing interest:** The authors declare that no competing interests exist.

## Introduction

Vertebrates exhibit great diversity in the number of segments that divide the skeleton, musculature, and nervous system of the body, along the rostral-caudal axis (*Richardson et al., 1998*). However, the developmental basis for this evolvability remains poorly understood. Segmentation of the body is established by paired blocks of mesodermal tissue, known as somites, that form periodically on either side of the developing embryo as it elongates posteriorly, in a process known as somitogenesis (*Morin-Kensicki et al., 2002*; *Gomez and Pourquié, 2009*; *Oates et al., 2012*). The total number of segments corresponds to the number of somites formed in the embryo, which is thought to be an emergent property of the morphogenesis of the pre-somitic mesoderm (PSM) and the dynamics of a molecular oscillator known as the segmentation clock (*Morin-Kensicki et al., 2002*; *Cooke and Zeeman, 1976*; *Gomez et al., 2008*; *Gomez and Pourquié, 2009*; *Harima et al., 2013*; *Schröter and Oates, 2010*). Here, we hypothesise that the evolvability of vertebrate segment number may be underpinned by developmental modularity of the segmentation clock and morphogenesis of the PSM (*Raff, 1996*), and that the potential of these two processes to evolve independently from one another may explain the diversity observed in vertebrate segment number.

The segmentation clock is a complex gene regulatory network thought to be driven by cell-autonomous oscillations of transcription factors in the Hes/Her family (*Palmeirim et al., 1997*; *Lewis, 2003*; *Harima et al., 2013*; *Schröter et al., 2012*; *Webb et al., 2016*). Noisy oscillations are synchronised across cells by delta-notch signalling (*Horikawa et al., 2006*; *Lewis, 2003*; *Venzin and Oates, 2020*), creating synchronous travelling waves of gene expression that pulse from the posterior to the anterior of the PSM. In the anterior PSM somite boundaries are patterned by the interaction of clock oscillations with a posterior-anterior decreasing gradient of FGF signalling that is thought to act as a 'wavefront', reading out the phase of the clock to create a spatially periodic pattern of somite boundaries (*Cooke and Zeeman, 1976*; *Soroldoni et al., 2014*; *Simsek and Özbudak, 2018*; *Simsek et al., 2023*). The clock synchronises the differentiation of PSM cells as they adopt somite fates at the anterior of the PSM (*Cooke and Zeeman, 1976*), and so controls both the tempo at which somites are formed and the anterior-posterior polarity of the somites (*Simsek et al., 2023*). Thus, clock synchrony controls the accuracy of somite patterning, and the frequency of clock oscillations determines the frequency of somite formation, which (together with the total duration of somitogenesis) determines the total number of somites formed.

Concurrent with somitogenesis, the PSM undergoes elongation. Comparative studies have shown that a diverse range of mechanisms are responsible for elongation of the PSM (*Gomez et al., 2008*; *Bénazéraf et al., 2010*; *Steventon et al., 2016*; *Mongera et al., 2018*; *Thomson et al., 2021*; *Michaut et al., 2022*). Importantly, the elongation dynamics of the PSM are thought to control the total duration of somitogenesis, which terminates once the PSM becomes sufficiently short in the AP direction (*Gomez et al., 2008*; *Gomez and Pourquié, 2009*; *Steventon et al., 2016*). Therefore, the total number of segments formed in the developing vertebrate is thought to be the product of both the dynamics of the clock (which controls the rate of somite formation) and the morphogenesis of the PSM, via its control on the total duration of somitogenesis.

Developmental modularity is a property of two or more developmental processes where their uncoupling in space or time permits their evolution independently of one another (*Raff, 1996*). This property is thought to give rise to increased phenotypic diversity by enhancing the evolvability of the system (*Raff, 1996*). In species examined thus far, such as the Corn snake *Pantherophis guttatus*, it appears that the evolution of segment number is driven by changes in both the dynamics of the clock and the elongation of the PSM (*Gomez et al., 2008*), and it is possible that independent evolution of these two processes is responsible for the high degree of diversity observed in vertebrate segment number. However, it is not obvious whether the two processes are modular, and if so, to what degree. Perturbing the clock's periodicity and function does not appear to affect elongation of the body axis (*Schröter and Oates, 2010*; *Lleras Forero et al., 2018*), so morphogenesis of the PSM is likely to be robust to changes in the segmentation clock through evolution. However, it is possible that many of the cellular- and tissue-level processes which drive PSM elongation could exert an effect on the dynamics of the clock.

For instance, cell rearrangements thought to drive elongation of the PSM in zebrafish (*Danio rerio*) (*Lawton et al., 2013*; *Mongera et al., 2018*) and in chicken (*Gallus gallus*) (*Bénazéraf et al., 2010*; *Michaut et al., 2022*), promote synchronisation of the segmentation clock (*Uriu et al., 2013*; *Uriu et al., 2010*; *Uriu and Morelli, 2014*; *Uriu et al., 2017*). Additionally, arrest of transcription during chromatin condensation is known to cause PSM cells to lag their clock expression relative to neighbours after cell division (*Horikawa et al., 2006*; *Delaune et al., 2012*), causing defects in clock synchrony (*Murray et al., 2013*) and suggesting clock synchrony may depend on the degree of proliferative growth in the PSM. The ingression of PSM progenitor cells from dorso-posterior and lateral donor tissues (*Kanki and Ho, 1997*; *Steventon et al., 2016*; *Xiong et al., 2020*) can also create clock asynchrony, as incoming progenitor cells do not appear to show clock gene expression (*Mara et al., 2007*) and thus may be asynchronous with their neighbours when they enter the PSM. Finally, tissue convergence movements associated with the elongation of the PSM (*Thomson et al., 2021*) have been proposed to negate the effect of random cell mixing and be deleterious for clock synchronisation (*Uriu and Morelli, 2014*). It is thus non-trivial to determine whether the duration and rate of somitogenesis are modular since, in order to do so, one must examine the effect of varying morphogenesis on the dynamics of the clock.

To do this, we use a computational approach to simulate clock dynamics and cell movements within the PSM, and study how the clock responds to changes in morphogenesis. We use this approach as it

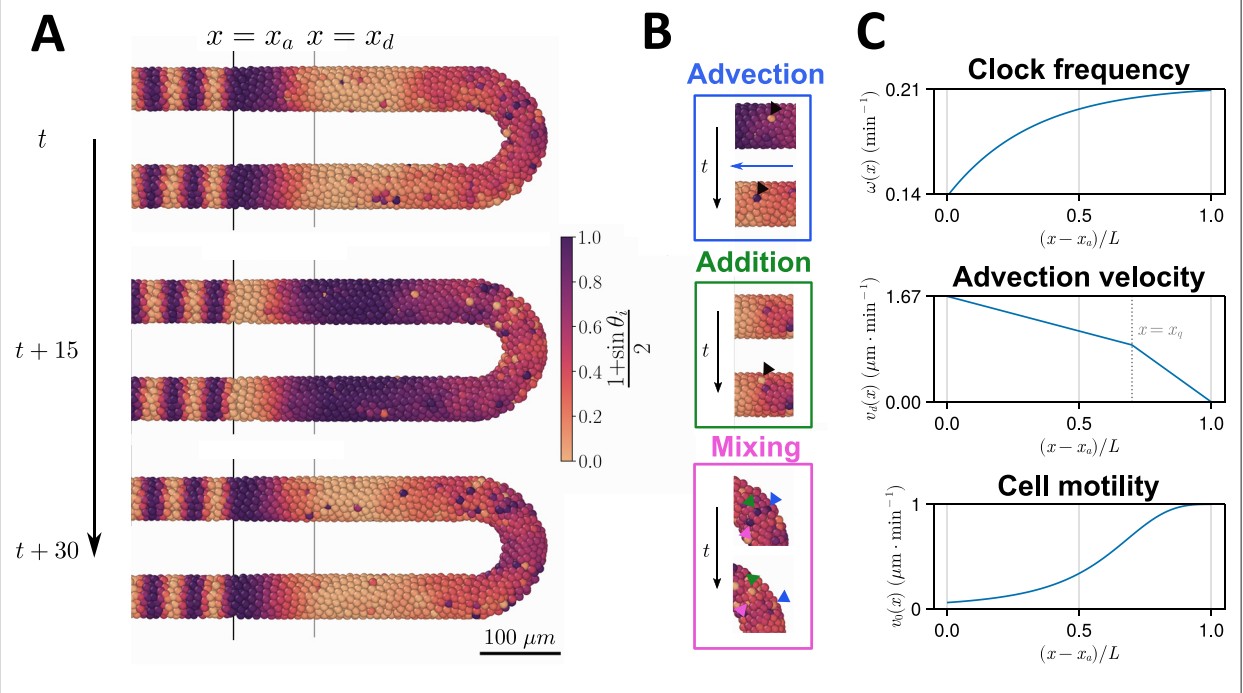

**Figure 1.** Computational model of the clock and the pre-somitic mesoderm (PSM). (**A**) Stills of a simulation of the model of *Uriu et al., 2021*. Kinematic phase ($\theta_i$) waves emerge in the posterior (right) and travel towards the tissue anterior (left, $x = x_a$), where phase is arrested. The model is parameterised to data from zebrafish, and accordingly the clock oscillates every 30 min. (**B**) Insets illustrating the key processes driving cell movements in the PSM within the model. Top: Cells advect towards the anterior of the tissue, simulating elongation of the PSM. Middle: New cells are added to replenish the loss of cellular material as cells advect towards the anterior. Bottom: Cells undergo motility-driven rearrangements. (**C**) Functions in the model describing (top) the intrinsic oscillation frequency, (middle) the advection velocity, (bottom) and the motility, of each cell depending on its normalised position along the anterior-posterior axis, $(x - x_a)/L$. Plots were generated using the parameters given by *Uriu et al., 2021*.

is much quicker and more flexible than experimentally manipulating morphogenesis in vivo. As we are constrained by computational complexity and cannot simulate all possible means by which the PSM can elongate (this would require simulation of the growth and dynamics of cells in surrounding tissues which exert forces on the PSM; *Xiong et al., 2020*), we limit our study to simulating regimes of cell movement and growth within the PSM that are known to be associated with, or causal in, PSM growth and elongation across vertebrates.

We use a previously established three-dimensional model of cell movements and clock dynamics within the zebrafish PSM (see *Figure 1A*; *Uriu et al., 2021*) and adapt this model to simulate various regimes of cell movement and growth. Briefly (a more detailed description of the model can be found in the Methods), the model assumes the PSM is in an inertial frame of reference, with elongation being encoded by the advection of cells towards the anterior PSM (the anterior limit of which is denoted $x = x_a$, see *Figure 1A*). Cells are modelled as point particles and are subject to random cell mixing, a cell-cell repulsion force that enforces volume exclusion of cells, and a boundary force confining the cells within a hollow horseshoe-shaped domain that reflects the geometry of the PSM. This model uses a phase-oscillator approach to describe segmentation clock dynamics and each cell is assigned a clock phase which is frozen when the cell exits the PSM at $x = x_a$, crossing the wavefront and patterning a segment (*Figure 1A*). To replenish cells lost at the anterior, new cells are added with random phase and random position to keep the tissue at a constant density (*Figure 1B*).

This model is appealing for several reasons, the first being that the model is three-dimensional and so can quantitatively recapture the rates of cell mixing that we observe within the PSM (*Uriu et al., 2017*; *Uriu et al., 2021*). The second is that the segmentation clock is described by an abstract phase-oscillator, broadly agnostic to the molecular details which are known to vary between vertebrates (*Krol et al., 2011*). Finally, the model makes extensive use of experimentally derived parameters for cell movements, tissue dimension, and the phase-oscillator model of the clock in zebrafish (*Uriu et al.,*

*2021*). This allows us to make quantitative predictions as to how the zebrafish segmentation clock would react to changes in PSM morphogenesis.

Using this model we test the clock's response to changes in the position of cell ingression into the PSM, changes in the anterior-posterior profile of random cell motility, changes in the length and density of the PSM, and to mitosis. We find that clock synchrony and frequency are robust to these changes except when mitosis is introduced, where we find that clock synchrony is negatively impacted except in cases where mitosis is confined to the posterior PSM. We find that this robustness is underpinned by the tissue length and cell density in the model, as well as the rate of cell mixing and strength of clock phase coupling. Together, these results suggest that segmentation clock dynamics and PSM morphogenesis are modular components of somitogenesis that can evolve independently from one another, conferring evolvability and helping to explain the diversity in segment number across the vertebrates.

## Results

### Varying the position of cell ingression has minimal effect on clock dynamics

During somitogenesis new cells enter the PSM posteriorly via ingression from dorsal tissues (*Kanki and Ho, 1997*; *Banavar et al., 2021*). In the early stages of zebrafish somitogenesis, there is also a contribution of cells from lateral tissues (*Steventon et al., 2016*). As ingressing cells do not appear to express segmentation clock genes (*Mara et al., 2007*), the position at which cells ingress into the PSM can create challenges for clock patterning, as only in the 'off' phase of the clock will ingressing cells be in-phase with their neighbours. Therefore, continuous ingression of PSM progenitor cells is likely to create local asynchrony of oscillations where it occurs, and so varying the position of cell ingression may have an effect on clock dynamics at the wavefront.

To test this hypothesis, we alter how cells are added to the tissue in the model. To model the ingression of cells from dorsal tissues into the posterior tailbud (*Kanki and Ho, 1997*; *Banavar et al., 2021*), we define a density-dependent cell addition process where cells are added onto the dorsal surface of the posterior half-toroid subdomain (see *Figures 2A* and 9) if the average density in this subdomain falls below $\rho_0$, with a constant initial phase $\theta = 0$. To maintain equal density across the PSM, we found it necessary to add cells in the two cylindrical PSM subdomain as well, as when cells were only added to the posterior half-toroid subdomain we observed a loss in density in the two cylindrical subdomains (data not shown). To avoid bias we add cells at random positions if the average density falls below $\rho_0$. These cells are initialised with a random phase drawn from the uniform distribution $[0, 2\pi]$. To model the early stages of zebrafish somitogenesis where, concurrent with dorso-posterior ingression of cells, the PSM also experiences a contribution of cells from lateral tissues, we modify the above simulation so that cell density in the lateral cylindrical subdomain is maintained by the addition of cells with constant phase ($\theta = 0$) on the ventral surface of these two lateral subdomain (see *Figures 2A* and 9). These two ingression scenarios are termed 'dorso-posterior' ('DP') and 'dorso-posterior+lateral-ventral' ('DP+LV') ingression (*Figure 2A*), respectively, and for brevity we refer to them as 'DP' and 'DP+LV' in the rest of the text. We compare the effect of these scenarios of cell ingression with 'Random' ingression (*Figure 2A*), where cell density is maintained by the addition of cells at random positions within the tissue with random phase ($\theta \in [0, 2\pi]$).

We test these scenarios by initialising each simulation with a synchronous initial condition for the clock and simulating cell movements and clock dynamics in the presence of each cell ingression scenario for 1000 min (see Methods for further justification). After 1000 min we measure the synchrony ($r$) and mean frequency ($d\theta/dt$) of a cylindrical region of tissue at the left-hand PSM anterior, one cell diameter ($d_c$) in length (see Methods). We restrict our measurement to the PSM anterior as it is here that the clock patterns somites (*Simsek and Özbudak, 2018*; *Simsek et al., 2023*) and, thus, dynamics elsewhere in the tissue are unlikely to present a phenotype in the animal. Using this method, we find that DP ingression has no effect on synchrony at the PSM anterior after 1000 min when compared to Random ingression, but that DP+LV ingression creates a minor decrease in anterior synchrony (*Figure 2B*). However, in neither scenario is the anterior frequency changed when compared to Random ingression (*Figure 2C*). We see equivalent results when ingressing cells are initiated with a random phase (*Figure 2—figure supplement 1*). Kymographs of synchrony along the

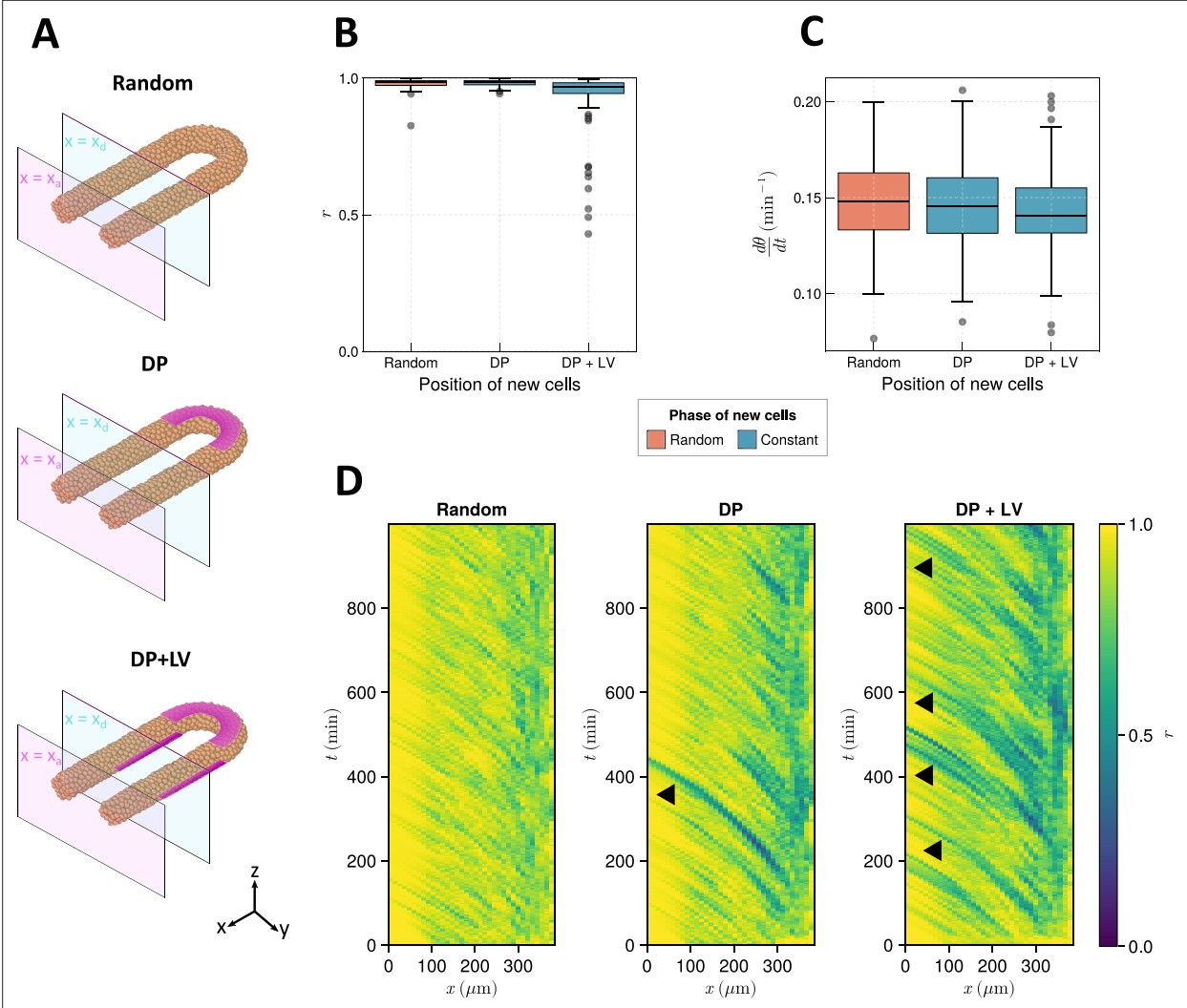

**Figure 2.** The effect of cell ingression position on clock frequency and synchrony. (**A**) Diagram highlighting the position of cell addition across the three ingression scenarios tested here. Magenta shading shows where cells are added onto the tissue surface. The pale magenta and blue planes respectively correspond to the anterior limit of the pre-somitic mesoderm (PSM), $x = x_a$, and the anterior limit of cell addition, $x = x_d$. In the 'Random' condition cells are added at random positions within the PSM posterior to the plane $x = x_d$, and accordingly no magenta surface is shown. Similarly in the dorso-posterior case ('DP') cells are added at random positions in the two lateral cylinders to maintain density, and no magenta surface is shown there. In the dorso-posterior+lateral-ventral ('DP+LV') case, cells are only added on the tissue surface at the positions shown by magenta shading. (**B**) Oscillation synchrony ($r$) at the PSM anterior after 1000 min, for the three tested scenarios of cell ingression. $N = 100$ simulations. (**C**) Mean frequency of oscillations for cells at the PSM anterior after 1000 min of simulation, for the three tested scenarios of cell ingression. $N = 100$ simulations. (**D**) Kymographs of synchrony along the $x$-axis on the left-hand side of the PSM for single simulations from each of the three scenarios of cell ingression tested. Black arrowheads highlight strongly asynchronous populations of cells being transported to the tissue anterior by advection.

The online version of this article includes the following figure supplement(s) for figure 2:

**Figure supplement 1.** Effect of varying cell ingression position with random initial phases.

anterior-posterior axis reveal that cell addition onto the tissue surface creates asynchronous populations of cells that travel towards the tissue anterior with cell advection (*Figure 2D*, black arrowheads). These are more frequent than in the DP case, likely explaining the lower anterior synchrony observed in the DP+LV simulations.

## The clock is robust to changes in mode of cell ingression regardless of the cell motility profile

As cell mixing promotes clock synchrony (***Uriu et al., 2013***; ***Uriu et al., 2010***; ***Uriu and Morelli, 2014***), and an anterior-posterior profile of increasing cell mixing is present in the PSM of zebrafish and chicken (***Kanki and Ho, 1997***; ***Bénazéraf et al., 2010***; ***Lawton et al., 2013***; ***Mongera et al., 2018***; ***Michaut et al., 2022***) (and presumably most vertebrate taxa), we speculated that the shape of this gradient may confer robustness to clock dynamics against cell ingression. For instance, a steep profile (e.g. as reported from the early somitogenesis stages of zebrafish; ***Mongera et al., 2018***) might confer robustness to lateral ingression as mixing is higher along a greater length of the PSM than a more graded profile (e.g. as reported from chicken embryos; ***Bénazéraf et al., 2010***; ***Michaut et al.,***

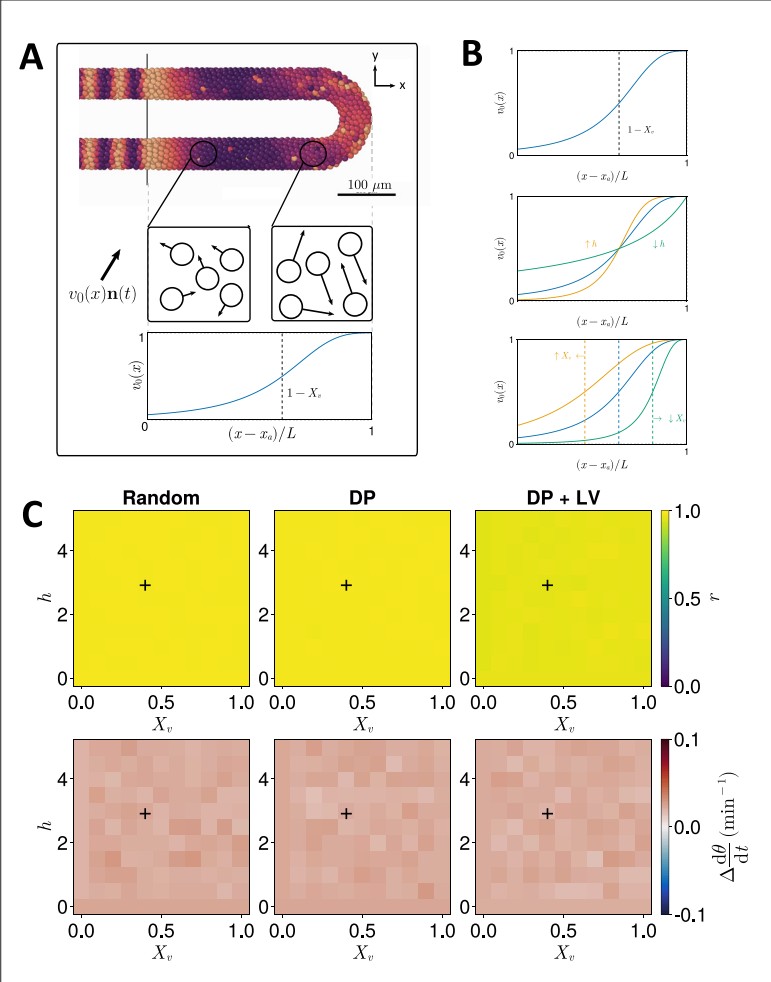

**Figure 3.** Effect of cell motility profile on clock frequency and synchrony. (**A**) Overview of how intrinsic cell motion is encoded in the model. Each cell is given a random direction vector $v_0(\mathbf{x}_i)\mathbf{n}_i(t)$ (black arrows) whose magnitude $v_0(\mathbf{x}_i)$ increases towards the pre-somitic mesoderm (PSM) posterior. (**B**) Magnitude of intrinsic cell motion $v_0(\mathbf{x})$ for the parameters used by ***Uriu et al., 2021*** (blue), and how the shape of the function can change when increasing (yellow) or decreasing (green) the inflexion point and curve steepness parameters, $X_v$ and $h$, respectively. (**C**) Clock synchrony $r$ (top) and difference from expected mean frequency $\Delta \mathrm{d}\theta/\mathrm{d}t$ (bottom) for different $v_0(\mathbf{x})$ specified by combinations of $X_v$ and $h$. The corresponding pixels display the synchrony or frequency at the PSM anterior after 1000 min of simulation using the specified motility profile, averaged across $N = 100$ simulations. A black + corresponds to the parameter pair used elsewhere in this paper, unless otherwise stated ($X_v = 0.4$, $h = 3$). Parameter ranges used are $X_v \in \{0, 0.1, 0.2, \ldots, 1\}$ and $h \in \{0, 0.5, 1, \ldots, 5\}$.

The online version of this article includes the following figure supplement(s) for figure 3:

**Figure supplement 1.** Clock synchrony along the x-axis for varying profiles of cell motility.

*2022*). Therefore, the shape of the cell motility profile in the PSM may constrain the evolution of cell ingression position and vice versa.

To test this, we systematically enumerated the steepness ($h$) and inflexion point ($X_v$) of the function $v_0(x)$ (*Figure 3B*), which controls the speed of random cell movement in the PSM (see Methods, *Figure 3A*). For each parameter pair we performed $N = 100$ simulations and recorded the median anterior synchrony and median anterior mean frequency ($d\theta/dt$, see Methods) after 1000 min. The results for each ingression scenario are shown in *Figure 3C*. We find no trend in median anterior synchrony and frequency across different parameter pairs for Random ingression, DP ingression, or DP+LV ingression (*Figure 3C*). Similar to before, results for DP ingression are indistinguishable from Random ingression. For DP+LV ingression, anterior synchrony is lower across all parameter sets than DP ingression or Random ingression (*Figure 3C*) but the effect is still very minor in this case. We do not observe any $X_v$- or $h$-dependent trend when we examine synchrony along the anterior-posterior axis (*Figure 3—figure supplement 1*). We therefore predict that, at least with respect to effects on the zebrafish segmentation clock, the evolution of the motility profile and the position of cell ingression are not constrained by one another.

## Length and density of the PSM confer robustness to the clock

The PSM is a transient tissue that shrinks in length over time. While this feature is conserved across vertebrate species, the absolute length and rate of shrinkage are known to vary between taxa (*Gomez et al., 2008*; *Steventon et al., 2016*; *Thomson et al., 2021*). We postulate that PSM length is an important factor in determining the clock's response to different morphogenetic modes; e.g., intuitively cells will have more time to synchronise oscillations with their neighbours before reaching the PSM anterior in a longer PSM, or if cell ingression is spread out along a longer tissue the degree of asynchrony imparted by ingression will be lessened. Similarly, we predict that the density of the tissue confers robustness to clock dynamics as increasing the average number of neighbours for each cell may act to 'correct' against clock noise. Overall, tissue density has been observed to increase over time in zebrafish (*Thomson et al., 2021*) and has been observed to vary in space in chicken, with density decreasing towards the PSM posterior (*Bénazéraf et al., 2010*; *Michaut et al., 2022*).

We thus sought to explore whether the length and density of the PSM confer robustness to changing morphogenesis. We test this by co-varying the position of cell ingression (*Figure 2A*) with either tissue density or length. When varying the average tissue density $\rho_0$ and studying model dynamics after 1000 min we see that increasing cell density marginally increases anterior synchrony (*Figure 4B*) and decreases the variability in frequency at the PSM anterior (*Figure 4C*). As expected, the number of neighbours for each cell is increased (*Figure 4—figure supplement 1*). However, the increase in density does not appear to be sufficient to 'correct' the decrease in synchrony in the DP+LV case (*Figure 4B*). Increasing the length of the tissue does however appear to 'correct' this decrease (*Figure 4E*). Here, the PSM length $L$ is increased but the anterior limit of cell addition $x_d$ is held constant at $x_d = 100\ \mu m$. Anterior synchrony after 1000 min positively correlates with $L$, and decreasing $L$ reveals a decrease in anterior synchrony for DP ingression (*Figure 4E*). No obvious trend for mean anterior frequency is observed (*Figure 4F*). Kymographs of synchrony along the $x$-axis suggest that while much of this correlation is driven by cells having more time to synchronise their neighbours with increasing $L$, a significant part of this may be due to cell addition occurring along a longer domain, and in so doing 'diluting' the noise-giving effect of cell ingression (*Figure 4—figure supplement 2*). To control for this, we restrict cell addition to only occur along a fixed length of the tissue with $x_d = L - R - r_T - 100$ (where $R$ represents the major radius of the half-toroid subdomain and $r_T$ the radius of the PSM tube, see Figure 9), and vary the length of the tissue $L$. We observe similar results as to before, with synchrony positively correlating with $L$ and no clear trend for clock frequency (*Figure 4—figure supplement 3*). Kymographs of synchrony along the $x$-axis confirm that increasing $L$ increases the effective time cells have to synchronise oscillations with their neighbours (*Figure 4—figure supplement 4*).

We note that when either one of density ($\rho_0$) or PSM length ($L$) increases, the differences in synchrony between the three scenarios of cell ingression become less pronounced (*Figure 4B and E*). This suggests that density and length are capable of independently conferring robustness to the segmentation clock against changes in morphogenesis, however, co-varying density and length suggests that the relative effect of length is stronger (*Figure 4—figure supplement 5*). In zebrafish

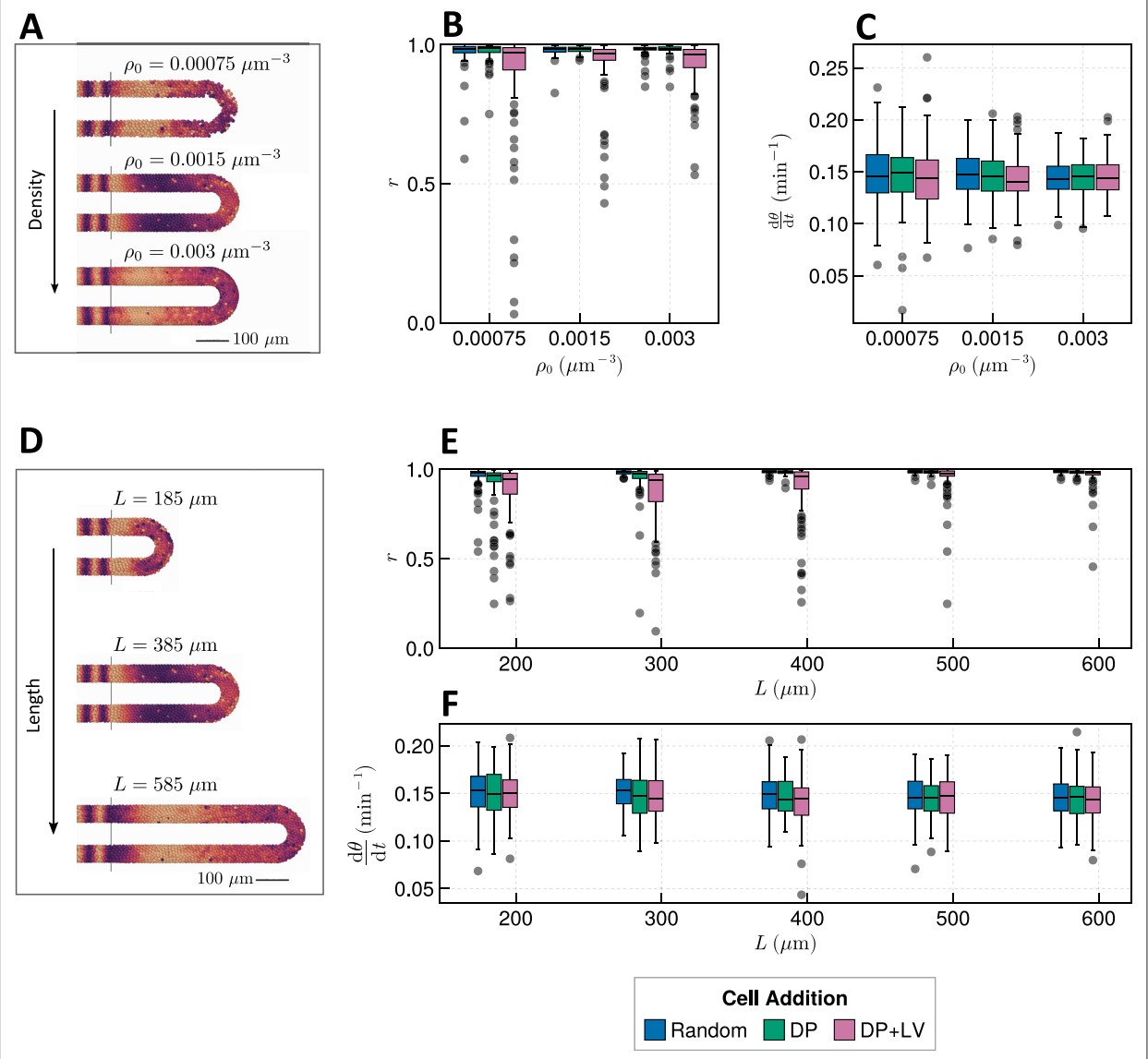

**Figure 4.** Effect of tissue density and length. (**A**) Stills from exemplar simulations illustrating the impact of changes in tissue density $\rho_0$. (**B**) Anterior synchrony after 1000 min for changing tissue density $\rho_0$ and varying position of cell ingression. $N = 100$ simulations. (**C**) Anterior mean frequency after 1000 min for changing tissue density $\rho_0$ and varying position of cell ingression. $N = 100$ simulations. (**D**) Stills from exemplar simulations illustrating changes in tissue length. (**E**) Anterior synchrony after 1000 min for changing tissue length $L$ and varying position of cell ingression. $N = 100$ simulations. (**F**) Anterior mean frequency after 1000 min for changing tissue length $L$ and varying position of cell ingression. $N = 100$ simulations.

The online version of this article includes the following figure supplement(s) for figure 4:

**Figure supplement 1.** Distribution of the number of neighbours for each cell when varying the density ($\rho_0$) of the tissue.

**Figure supplement 2.** Synchrony dynamics for varying tissue length $L$.

**Figure supplement 3.** Anterior synchrony for varying tissue length $L$, with varying anterior limit of cell addition $x_d$.

**Figure supplement 4.** Synchrony dynamics for varying tissue length $L$, with varying anterior limit of cell addition $x_d$.

**Figure supplement 5.** Clock dynamics for co-varying density $\rho_0$ and length $L$.

density and length are known to co-vary after the 16-somite stage (ss), with the PSM increasing in density as it shrinks in length (*Thomson et al., 2021*). We would expect decreasing the length of the PSM to decrease the clock's robustness to noise, but it is possible that the concurrent increase in density is sufficient to counteract this and preserve clock dynamics. As the dynamics of somitogenesis are consistent across individuals at these late stages (*Schröter et al., 2008*), the dynamics of the

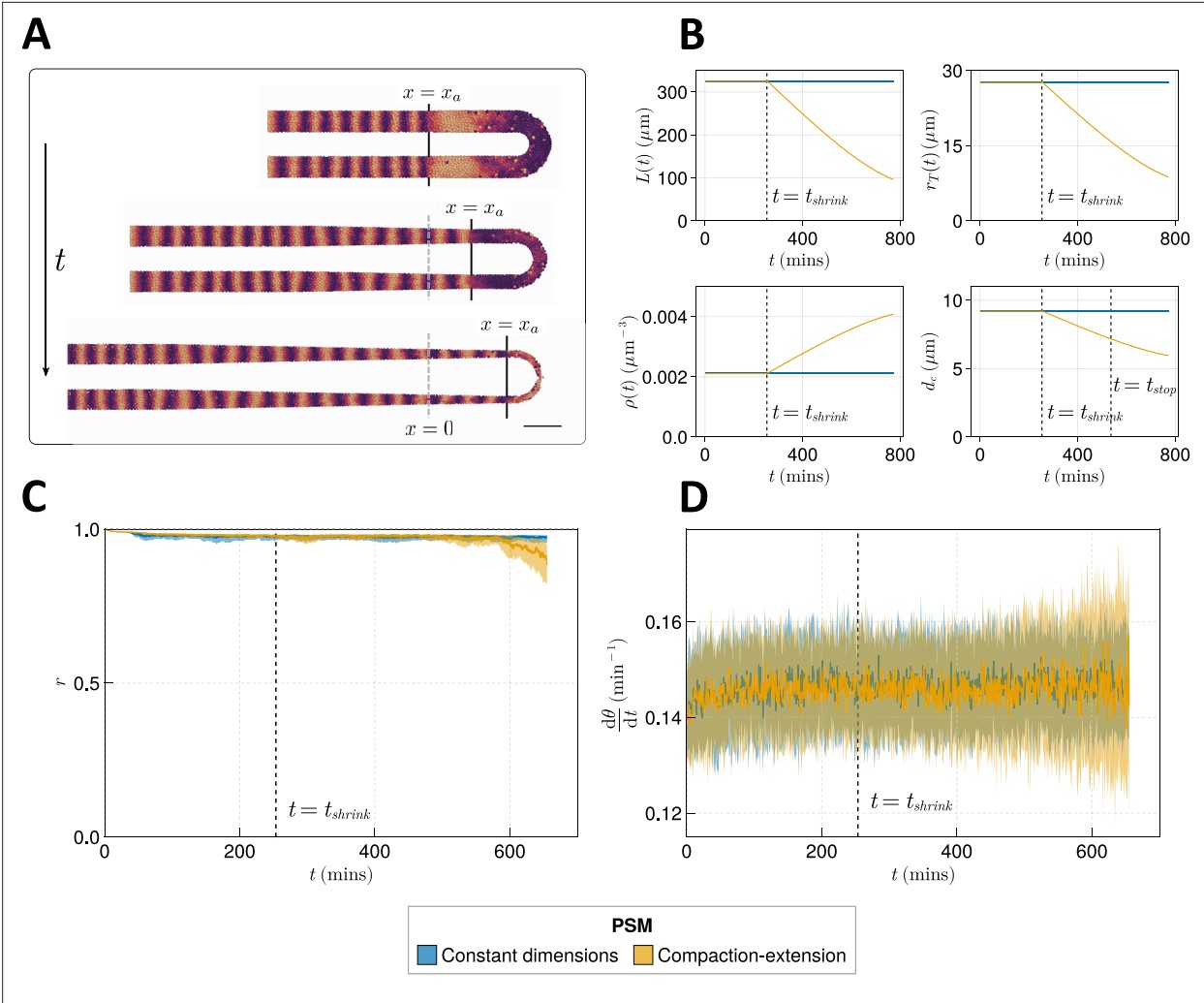

**Figure 5.** Effect of compaction-extension on clock synchrony and frequency. (**A**) Snapshots of an exemplar simulation showing how the pre-somitic mesoderm (PSM) shrinks in length and diameter as time progresses. A 100 μm scalebar is shown in the lower right-hand side of the figure. (**B**) Functions for PSM length $L$, radius $r_T$, density $\rho$, and cell diameter $d_c$, derived from *Thomson et al., 2021* (yellow), and the constant functions (blue) with which the effect of these functions is compared. (**C**) Anterior synchrony and (**D**) mean anterior frequency over time, for $N = 100$ simulations. The solid line indicates the median and the inter-quartile range is given by a shaded band either side of this line. Blue shows simulations where the tissue does not undergo compaction-extension after $t_{shrink}$, and yellow shows simulations where after $t_{shrink}$ the tissue undergoes compaction-extension according to the functions shown in **B**. Results are plotted until the time at which at least one of the replicate simulations encounters a gap in the tissue at the tissue anterior (see Methods).

The online version of this article includes the following figure supplement(s) for figure 5:

**Figure supplement 1.** Zebrafish somitogenesis, after *Schröter et al., 2008*.

**Figure supplement 2.** Comparison of a 'stepwise' function for decreasing cell diameter $d_c$ against a continuously decreasing function.

**Figure supplement 3.** Effect of starting tissue density $d_0$ on anterior synchrony and frequency over time, for a compacting tissue.

zebrafish segmentation clock are likely unaffected by compaction-extension of the PSM. It is not clear however if this can be explained by the concurrent changes in PSM length and density that occur during compaction-extension or if clock properties such as phase coupling need to vary in order to maintain robust clock dynamics.

Here, we simulate compaction-extension by shrinking the PSM radius $r$ and length $L$, while increasing its density $\rho$ over time, using published rates and initial conditions for these values (*Figure 5A and B*, Methods) (*Thomson et al., 2021*). We also decrease the cell diameter $d_c$, as has been observed in zebrafish (*Thomson et al., 2021*), as we expect this to be important in determining clock dynamics by changing the number of neighbouring cells that a given cell can couple its phase to.

As these changes are dynamic, we depart from previous methodology for measuring clock dynamics and plot the anterior synchrony and frequency over time for $N = 100$ simulations. We also no longer simulate for 1000 min as in previous simulations, rather terminating the simulation when the length of the tissue $L(t)$ is less than $x_d$ (i.e. when $x_a(t) + x_d \geq X_c + R + r_T(t)$). We impose this constraint so as to prevent asynchrony induced by random addition of new cells biasing our results. It is important to note that we therefore only simulate for a part of the total duration of somitogenesis in zebrafish, and that there may be dynamics in the terminal stages of somitogenesis not captured by this model.

The anterior synchrony and mean anterior frequency over time for $N = 100$ simulations of compaction-extension are shown in *Figure 5C and D*, respectively. They are compared with simulations that do not shrink the PSM and maintain constant $L$, $r_T$, $d_c$, and $\rho$. We find that towards the end of these simulations the clock experiences more noise and fluctuations in synchrony and frequency (*Figure 5C and D*) than the non-shrinking equivalent simulations, however, for the majority of the simulation the dynamics of the compacting tissue are broadly comparable with those of the non-compacting tissue, but with synchrony minorly decreasing and frequency becoming noisier towards the end of the simulation (*Figure 5C and D*). As model parameters that are fixed throughout the simulation, such as advection velocity or intrinsic frequency, are thought to change towards the end of somitogenesis (*Schröter et al., 2008*; *Uriu et al., 2021*), it is possible that for the latter stages of somitogenesis the results of our simulations are inaccurate. Data for the terminal stages of somitogenesis are extremely limited and we cannot make estimates for model parameters at this stage. Therefore, we conclude that for much of somitogenesis in zebrafish the clock is not majorly impacted by the compaction-extension of the tissue.

Beyond dynamic changes in density, we also attempted to encode a spatial gradient of decreasing cell density towards the posterior, as reported from chicken (*Bénazéraf et al., 2010*; *Michaut et al., 2022*). However, we could not generate a gradient that quantitatively recaptures that reported by *Bénazéraf et al., 2010* (data not shown). Based on our results for changing average cell density $\rho_0$ (*Figure 4B and C*), it is highly probable that this gradient would affect clock dynamics. We suggest that another model, such as one encoding cell-cell adhesion, may be a more powerful tool for exploring the effect of this gradient and dynamics at the terminal stages of somitogenesis.

## Clock arrest during cell division creates asynchrony

During M-phase of the cell cycle, segmentation clock gene expression in zebrafish is known to arrest (*Horikawa et al., 2006*), causing a lag in nascent daughter cells relative to their neighbours (*Delaune et al., 2012*). If cell divisions are asynchronous, as appears to be the case in zebrafish (*Kanki and Ho, 1997*; *Steventon et al., 2016*), then such lags can create asynchrony of oscillations in the tissue (*Murray et al., 2013*).

As previous models of mitosis and the segmentation clock have been two-dimensional and therefore may not quantitatively recapture the cell mixing and geometry present in zebrafish (*Murray et al., 2013*; *Murray et al., 2019*), we sought to investigate the effect of mitosis in the presence of the more accurate three-dimensional geometry and cell mixing. To do so, we simulated cell division by assigning each cell a cell cycle phase $\tau$ that increases at a constant rate of $1 \text{ min}^{-1}$, and spawning a new cell adjacent to that cell once that its $\tau \geq T_G + T_M$, where $T_G$ denotes the total time in minutes to complete G1, S, and G2 phases of the cell cycle, and $T_M$ denotes the time spent in M-phase, in minutes. Once a daughter cell is generated, the $\tau$ for both cells is reset to zero. As daughter cells share the same clock phase after division in vivo (*Delaune et al., 2012*), the daughter cell shares the same clock phase $\theta$ as its sibling. To simulate the arrest of the clock during M-phase, if $\tau \in [T_G, T_G + T_M]$ oscillations cease, i.e., $\mathrm{d}\theta_i/\mathrm{d}t = 0$. For simplicity, we allow neighbouring cells to couple their phase to that of a neighbour with arrested phase, as transcriptional arrest is presumably equivalent to the 'off' phase of the segmentation clock. For further details, we refer the reader to the Methods section below.

To avoid measuring trivial changes in frequency and synchrony due to arrest of the clock, we measure synchrony and frequency of the clock only in cells where $\tau < T_G$. Therefore, our results represent a lower bound on the effect of mitosis on the clock. Using the estimated values of $T_G$ and $T_M$ for zebrafish (see Methods for calculation), we find that the presence of cell division creates asynchrony as reported previously (*Murray et al., 2013*) and has no identifiable effect on frequency (*Figure 6B and C*). Fixing $T_G + T_M = 187.5$ min and varying $T_M$ shows a $T_M$-dependent trend where

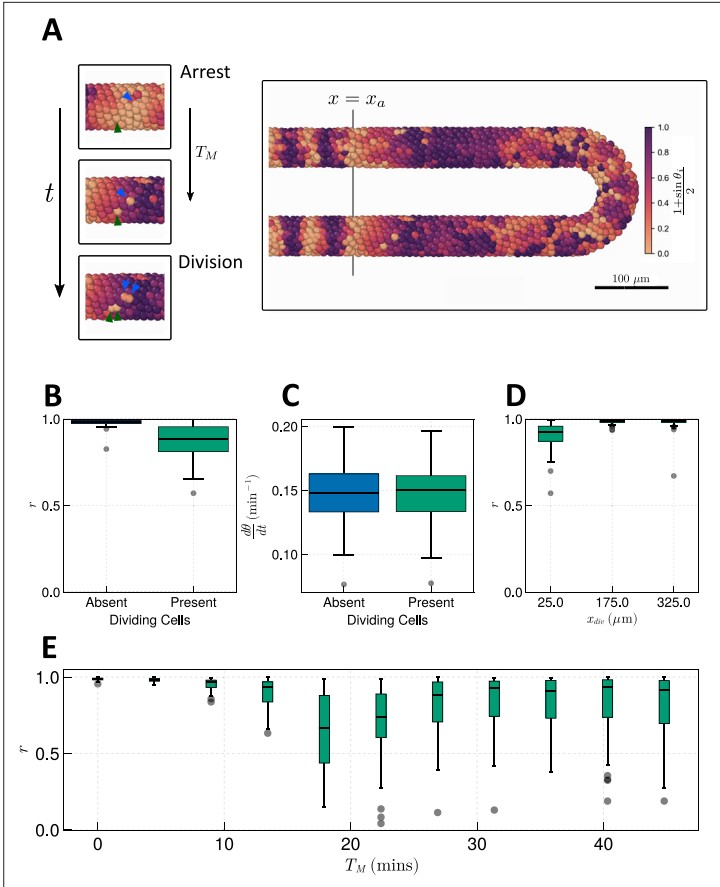

**Figure 6.** Effect of cell division on clock frequency and synchrony. (**A**) Diagram showing how during cell division, clock phase $\theta$ arrests, causing a cell to fall out of phase with its neighbour. On the right-hand side a still from a simulation for $T_M = 15$ min is shown, illustrating how this creates asynchrony of oscillations. The effect of cell division on anterior synchrony and frequency is shown in figures **B and C**, respectively, for $T_M = 15$ min after 1000 min. $N = 100$. (**D**) The effect on anterior synchrony when division is restricted to only occur posterior to $x = x_{div}$, for $T_M = 15$ min. $N = 100$. (**E**) The effect on anterior synchrony after 1000 min when $T_M$ is varied. In each case, the total length of the cell cycle is maintained at a constant length, i.e., $T_M + T_G = 187.5$ min. $N = 100$. To rule out trivial changes in synchrony and frequency, in all analysis here we restrict measurement to non-dividing cells, i.e., cells such that $\tau \in [0, T_G)$.

The online version of this article includes the following figure supplement(s) for figure 6:

**Figure supplement 1.** The effect of cell division on frequency.

**Figure supplement 2.** Effect of cell division where clock coupling is forbidden during M-phase.

**Figure supplement 3.** Spatiotemporal constraints on $T_M$.

synchrony decreases with increasing $T_M$ (**Figure 6E**). After $T_M = 15$ min the synchrony recovers slightly but remains noisy (**Figure 6E**). No noticeable trend is observed for the frequency (**Figure 6—figure supplement 1A**). We observe similar results when neighbouring cells are forbidden from coupling their phase to neighbours in M-phase (**Figure 6—figure supplement 2**).

In previous simulations (**Figure 6**), cell division occurs up to the wavefront where segments are pre-patterned ($x = x_a$). However, if division is confined to the posterior (**Bouldin et al., 2014**), cells may have time to re-synchronise before reaching the wavefront. To test whether this is possible within the timescales of zebrafish cell movement, division, and clock dynamics, we define a point in space $x = x_a + x_{div}$ anterior to which (i.e. for $x < x_a + x_{div}$) we halt cell cycle progression by fixing $\tau = 0$ min in all cells. Performing simulations with $x_{div} = 25\,\mu m$, $175\,\mu m$, $325\,\mu m$, we see that while for $x_{div} = 25\,\mu m$ we observe defects in synchrony, for $x_{div} = 175\,\mu m$, $x_{div} = 325\,\mu m$, synchrony is rescued (**Figure 6D**). This suggests that there exists some point along the AP axis of the PSM, posterior to which cell division

can occur without affecting the anterior dynamics of the clock. To try and resolve the value of this point and how it depends on the length of M-phase $T_M$, we systematically enumerated values of $x_{div}$ from the tissue anterior to the tissue posterior and $T_M$ across the values tested in **Figure 6E**. We find that for $T_M < 15$ min, division can occur up to the tissue anterior without a major defect in synchrony (**Figure 6—figure supplement 3A**). For $T_M > 15$ min, simulations with division show high synchrony at the anterior although not so high as those with $T_M < 15$ min, where the value of $x_{div}$ such that clock synchrony is preserved is displaced to the posterior (**Figure 6—figure supplement 3A**). However, only for $T_M = 15$ min is this value majorly displaced to the posterior, with $x_{div} \approx 150\,\mu$m being the anterior-most value of $x_{div}$ such that synchrony is preserved ($r \approx 1$) (**Figure 6—figure supplement 3A**). This result is perhaps intuitive, as the parameters in the model set the clock period to $2\pi/\omega_0 = 30$ min, and a clock arrest lasting $T_M = 15$ min would be sufficient to move a cell into antiphase relative to its neighbours. Indeed, we see that mitosis is most deleterious to synchrony when M-phase lasts approximately half the length of a clock cycle (**Figure 6—figure supplement 3B**), however, for long clock cycles this relationship becomes nonlinear and the most deleterious values of $T_M$ lie between the length of one or two clock cycles (**Figure 6—figure supplement 3B**).

## Zebrafish clock coupling and cell mixing confer robustness to changes in cell ingression

Across many of the morphogenetic scenarios tested here, we have observed a high degree of robustness in clock dynamics with respect to changing cell movements and processes. As shown above, some of this robustness can be attributed to the choice of tissue length $L$ in simulations and choices of the parameters $v_a$ and $v_p$ governing the advection velocity of cells. However, we also expect that the rate of cell mixing (**Uriu et al., 2013**; **Uriu et al., 2017**; **Uriu et al., 2010**) and the strength of clock coupling confer robustness too. Within the present model, the global rate of cell mixing is controlled by the parameter $v_s$ and the clock coupling strength is denoted by $\kappa$. Notably, the values taken for these parameters have been experimentally measured in zebrafish (**Riedel-Kruse et al., 2007**; **Herrgen et al., 2010**; **Uriu et al., 2017**; **Uriu et al., 2021**). Therefore, we were interested in where these parameters lie within the space of parameter pairs that generate clock dynamics robust to changes in morphogenesis - do these results reflect something unique about the zebrafish parameters or can many ($\kappa$, $v_s$) pairs confer robustness?

To determine this, we take changing cell ingression as an example of varying morphogenesis and for each ingression scenario, systematically enumerate the parameters $v_s$ and $\kappa$, studying the anterior

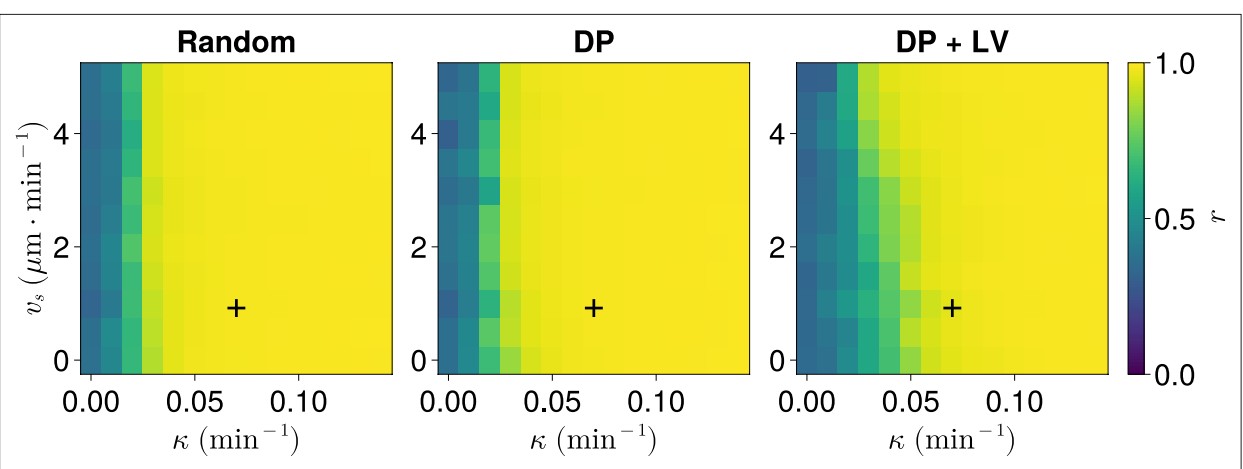

**Figure 7.** Synchrony for varying coupling strength $\kappa$ and magnitude of intrinsic cell motion. $v_s$. Anterior synchrony after 1000 min, for varying maximum magnitude of intrinsic cell motion $v_s$ and clock phase coupling strength $\kappa$, for three different scenarios of cell ingression. Each pixel corresponds to the median value of anterior synchrony across $N = 100$ simulations. A black + marks the experimental values for zebrafish, $\kappa = 0.07$ min$^{-1}$, $v_s = 1\,\mu$m $\cdot$ min$^{-1}$, derived in **Riedel-Kruse et al., 2007**, and **Uriu et al., 2017**, respectively, that are used elsewhere in this paper. All other parameters are held constant at their normal values (see **Table 1**).

The online version of this article includes the following figure supplement(s) for figure 7:

**Figure supplement 1.** Synchrony for varying coupling strength $\kappa$ and tissue length $L$.

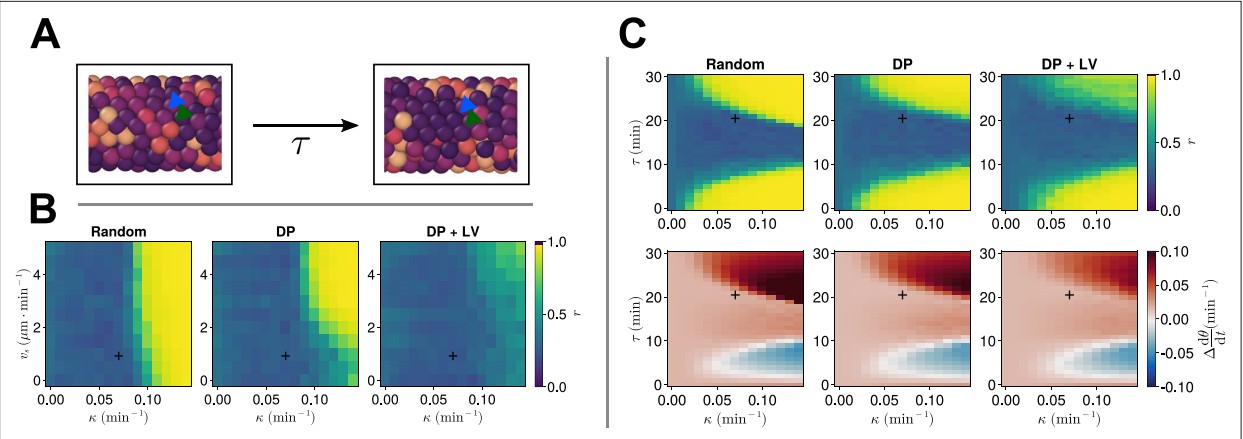

**Figure 8.** Clock dynamics in the presence of a coupling delay between cells. (**A**) Diagram illustrating phase coupling with delay. Cells couple their phase to the phase of their neighbours $\tau$ min ago. (**B**) Anterior synchrony in the presence of coupling delay ($\tau = 21$ min) after 1000 min, for varying maximum magnitude of intrinsic cell motion $v_s$ and clock phase coupling strength $\kappa$, for three different scenarios of cell ingression. Each pixel corresponds to the median value of anterior synchrony across $N = 100$ simulations. A black + marks the experimental values for zebrafish, $\kappa = 0.07$ min$^{-1}$, $v_s = 1$ μm · min$^{-1}$, derived in *Riedel-Kruse et al., 2007*, and *Uriu et al., 2017*, respectively, that are used elsewhere in this paper. All other parameters are held constant at their normal values (see *Table 1*). (**C**) Anterior synchrony (top) $r$ after 1000 min, for varying phase coupling delay $\tau$ and clock phase coupling strength $\kappa$, for three different scenarios of cell ingression. Each pixel corresponds to the median value of anterior synchrony across $N = 100$ simulations. Bottom: Difference from expected mean frequency ($\Delta d\theta/dt$) at the pre-somitic mesoderm (PSM) anterior after 1000 min, for three different scenarios of cell ingression. Each pixel corresponds to the mean value across $N = 100$ simulations. A black + marks the experimental values for zebrafish, $\kappa = 0.07$ min$^{-1}$, $\tau = 21$ min, derived in *Riedel-Kruse et al., 2007*, and *Herrgen et al., 2010*, respectively. All other parameters are held constant at their normal values (see *Table 1*).

The online version of this article includes the following figure supplement(s) for figure 8:

**Figure supplement 1.** Frequency in the presence of coupling delay, for varying coupling strength $\kappa$ and magnitude of intrinsic cell motion.

**Figure supplement 2.** Clock dynamics in the presence of a coupling delay, for varying tissue length $L$.

**Figure supplement 3.** Clock dynamics in the presence of a coupling delay, for varying tissue length $L$ and anterior limit of cell addition $x_d$.

synchrony after 1000 min with $N$=100 replicates per parameter pair. This reveals a threshold curve of ($\kappa$, $v_s$) pairs below which the clock begins to exhibit asynchrony (*Figure 7*). Comparing across the three scenarios of cell ingression, we see that the position of this curve changes with the cell ingression scenario (*Figure 7*), indicating that the space of ($\kappa$, $v_s$) pairs that achieve clock synchrony is constrained by the cell ingression mode present within the tissue. Notably, the experimentally derived parameter pair ($\kappa = 0.07$ min$^{-1}$, $v_s = 1$ μm · min$^{-1}$) lies very close to the threshold for the DP+LV case of cell ingression in the $\kappa$ direction, suggesting that the robustness we have observed in response to changes in cell ingression is sensitive to the choice of $\kappa$. However, in each case our experimental choices for $\kappa$ and $v_s$ lie above the threshold, implying that at least part of the robustness we observed with respect to cell ingression is due to the experimental values for zebrafish being sufficient to confer this robustness.

Finally, we test if this robustness holds in the presence of a phase coupling delay between cells. We add to our simulations the time taken for information about the transcriptional state of a cell's neighbour to be transduced by cell-cell signalling and impact the cell's transcriptional profile. Such delays are important to consider as they can be significant for clock synchrony and frequency (*Lewis, 2003*; *Morelli et al., 2009*), however, they significantly increase the computational complexity of such a model. The value of the cell-cell phase coupling delay has been estimated to be 21 min in zebrafish (*Herrgen et al., 2010*), and so to test the effect of delays we repeat the analysis above (*Figure 7*) in the presence of a cell-cell coupling delay ($\tau$) of 21 min.

We observe that, for $\tau = 21$ min, oscillations are asynchronous in all three scenarios of cell ingression with the experimentally measured values of $\kappa$ and $v_s$, however, for larger values of $\kappa$ (and, to a lesser extent, $v_s$), we observe synchronous oscillations (*Figure 8B*). Notably, we cannot achieve synchronous oscillations for any of the chosen parameter pairs for the DP+LV scenario of cell ingression (*Figure 8B*), suggesting that ingression of progenitor cells is a far greater source of noise for the clock in the presence of coupling delay than in the instantaneous coupling case. To test this, we varied

the length of the PSM $L$ and either restricted the addition of cells to occur within a region of length $100 + R + r_T$ at the tissue posterior or allowed cell addition to occur at any point in the tissue posterior to $x_d = 100$ μm. In either case the rate of cell addition is assumed to be constant and thus in the latter case we expect the longer tissue domain to 'dilute' the noise effect of cell ingression. We observe no difference between either scenario and find no value of $L$ such that synchronous oscillations are recovered for $\tau = 21$ min (*Figure 8—figure supplements 2 and 3*).

To explore how the co-evolution of the clock coupling strength $\kappa$ and the coupling delay $\tau$ might constrain the position of cell ingression, we systematically enumerated values of $\kappa$ as before while co-varying $\tau$, and analysed clock synchrony across the three different scenarios of cell ingression (*Figure 8C*). In agreement with previous modelling work (*Morelli et al., 2009*; *Herrgen et al., 2010*), we see that synchrony depends on the value of $\tau$ in a repeating manner, with values of $\tau$ closest to half of a clock cycle (~ 30 min) being more asynchronous (*Figure 8C*). These results also predicted that frequency decreases with increasing $\tau$ within regions of stability (*Morelli et al., 2009*; *Herrgen et al., 2010*). We see evidence of this when we examine the collective frequency at the PSM anterior, where within each region of synchrony the frequency decreases as $\tau$ increases (*Figure 8C*). We note that for small values of $\tau$ the DP+LV scenario of cell ingression is synchronous, however, for larger values of $\tau$ oscillations are largely asynchronous, unlike the DP scenario of ingression (*Figure 8C*).

We note that, despite our model failing to achieve synchronous oscillations for $\tau = 21$ min, only relatively minor increases of $\tau$ are required to achieve synchrony in the Random and DP+LV scenarios of cell ingression (*Figure 8C*). This closely resembles the analytical results of *Herrgen et al., 2010*, who found that the measured delay $\tau = 21$ min lies towards the extreme end of values of $\tau$ for which stable oscillations can occur (*Herrgen et al., 2010*). Together these results suggest that the zebrafish segmentation clock might exist in a state of criticality and close to a region of parameter space where oscillations are asynchronous. Overall our results suggest that coupling strength and coupling delay strongly constrain the evolvability of cell ingression, and that robustness to morphogenesis is conferred only for specific combinations of these two parameters.

## Discussion

Here, we investigated whether morphogenesis of the PSM and the segmentation clock exhibit developmental modularity, as this could explain the high degree of evolvability observed in vertebrate segment number (*Raff, 1996*). We tested a broad range of PSM cellular behaviours and mechanisms that are associated with the elongation of the PSM across vertebrate species, to see if they had an effect on the dynamics of the segmentation clock. We predict that the dynamics of the zebrafish segmentation clock, specifically synchrony and frequency, are generally robust to changes in these mechanisms such as cell ingression, motility, and (under certain conditions) division. As PSM morphogenesis is independent of clock function (*Schröter and Oates, 2010*; *Lleras Forero et al., 2018*), our results suggest that the clock and morphogenesis of the PSM exhibit developmental modularity.

Our results suggest that a major determinant of this robustness is the clock coupling $\kappa$, which in zebrafish appears to be sufficiently strong to confer robustness to changes in cell ingression (*Figure 7*). While synchrony of the clock shows a dependence on cell mixing rate ($v_s$) and tissue length ($L$), the dependence of synchrony on these parameters is weak compared to its dependence on coupling strength (*Figure 7*, *Figure 4—figure supplement 1*), suggesting that this robustness may be more dependent on properties of the segmentation clock than those of the tissue. Due to a lack of quantitative understanding of PSM morphogenesis, it is difficult to predict whether the requirements for robustness imposed on cell mixing or tissue length and density constrain the space of possible morphogenetic processes that can be altered without perturbing the clock. However, our results highlight that these requirements are themselves constrained by the strength of clock phase coupling, so it is equally possible that vertebrate segmentation clocks have strong phase coupling generally, and these requirements on morphogenesis are sufficiently relaxed to not constrain the evolution of PSM elongation. Further work, investigating the evolution of phase coupling strength in vertebrates, and developing accurate quantitative models of PSM morphogenesis, will be necessary to determine this.

It is also worth noting that while in the model formulation $\kappa$ represents the strength of delta-notch signalling between cells, the experimental value for this parameter is measured at a tissue level, incorporating coupling effects from other pathways, e.g., cell mixing (*Riedel-Kruse et al., 2007*). It is therefore possible that the value of $\kappa$ measured by *Riedel-Kruse et al., 2007*, represents an overestimate

of the strength of delta-notch signalling and therefore our results present an overestimate of robustness to morphogenesis. However, our results appear less sensitive to cell mixing ($v_s$) than $\kappa$ (*Figure 7*), so we suggest that the contribution of cell mixing to the estimate of $\kappa$ may be minor. Furthermore, as the precise value of $r$ necessary for correct segment patterning is unknown, and as downstream processes in segmentation may act to correct any defects in clock patterning (*Naganathan et al., 2022*), somitogenesis may be more robust than we assume here. Therefore, we suggest that our findings are robust to any overestimates of $\kappa$ by incorporation of cell mixing.

Our results when considering a phase coupling delay between cells (*Figure 8*) suggest that evolution of cell ingression is constrained by both the clock coupling delay between cells and vice versa. This suggests that the robustness of the segmentation clock to changes in morphogenesis is mediated by the combination of the cell-cell coupling delay and the coupling strength of the clock. While little is known about what determines the coupling strength of the clock, coupling delay is thought to be at least in part controlled by the number of intermediary components involved in signal transduction (*Lewis, 2003*; *Yoshioka-Kobayashi et al., 2020*). Therefore, the morphogenesis of the PSM may constrain the ability of the segmentation clock to undergo developmental system drift, at least within the components of the delta-notch pathway.

We note that our model fails to recover synchrony for the experimentally measured values of $\kappa$ and $\tau$, and thus may not accurately model the in vivo condition in some way. It is possible that the way in which we simulate cell ingression introduces too much noise into the clock, for instance the rate of progenitor cell ingression may be much higher than that in vivo, and indeed we note that elongation is driven in vivo by a combination of processes, rather than solely ingression, as we implicitly assume here. If this is true, this implies that clock imposes a constraint on the degree to which ingression is capable of driving PSM elongation. This possibility is worthy of further study and is perhaps better suited to models where PSM elongation is not dependent on cell ingression, as we assume here.

Furthermore, as discussed above, we note that for relatively small increases in the value of $\tau$, e.g., for $\tau = 25$ min, it is possible to achieve synchronous oscillations in both the random and DP scenarios of cell ingression, and partially synchronous oscillations in the DP+LV scenario of cell ingression (*Figure 8C*). This could suggest that the biological parameters controlling delay and cell ingression exist in a state of criticality, i.e., lying close to a region of parameter space where dynamical behaviour abruptly changes or becomes chaotic. Such states can have important implications for evolvability of a biological system (*Kauffman, 1993*; *Verd et al., 2019*), however, it is not clear if such a state is functional in this case. Further study, perhaps with models unlike the mean-field approximation we assume here, may reveal what implication, if any, this has for the evolvability of PSM morphogenesis.

Our analysis is restricted to the modes of PSM morphogenesis that are known to occur in vertebrates, and we cannot rule out the possibility that the robustness we observe merely reflects evolution of mechanisms of PSM elongation under selective pressure to minimise their effect on the clock, and that the clock may not be robust to all possible modes of PSM elongation. Nevertheless, we argue that our results demonstrate there exists a qualitatively distinct set of processes (cell ingression, division, motility, and compaction-extension) that underpin PSM elongation and do not affect clock dynamics. As vertebrate species appear to employ a combination of these processes to varying degrees (*Bénazéraf et al., 2010*; *Steventon et al., 2016*; *Mongera et al., 2018*; *Xiong et al., 2020*; *Thomson et al., 2021*), it is possible that evolution alters each of these processes independently to generate diversity in the elongation dynamics of the PSM (*Gomez et al., 2008*; *Steventon et al., 2016*). Therefore, this relatively small set of processes may be sufficient to explain the high degree of evolvability in vertebrate segment number.

As clock dynamics appear to be robust to changes in morphogenesis, we suggest that the clock and morphogenesis of the PSM comprise two developmental modules within zebrafish, and that this is true more generally whenever the phase coupling of the clock is sufficiently strong, cell mixing sufficiently rapid, or the PSM sufficiently dense and long. We suggest that this may have contributed to the evolution of broad diversity in vertebrate segment number (*Richardson et al., 1998*) and be a present source of evolvability in vertebrates. We note that the evolution of modularity is an active area of research (*Wagner and Altenberg, 1996*; *Wagner et al., 2007*) and suggest that studying the evolution of delta-notch signalling in vertebrates, and the evolutionary origin of the PSM and somitogenesis, may be an illuminating model paradigm for this field. We predict that the modularity of the clock and morphogenesis of the PSM permits a synergy between these two processes that heightens

the evolvability of segment number in vertebrates, and that this may be responsible for the high diversity we observe in this trait (*Richardson et al., 1998*).

Our results suggest that the zebrafish segmentation clock is robust to most changes in PSM morphogenesis, and that this robustness is determined by the strength of clock phase coupling and the phase coupling delay, as well as morphogenetic parameters such as the rate of cell mixing, and the length and density of the PSM. This suggests that, at least in the regions of parameter space where these conditions for robustness hold, PSM morphogenesis is not constrained by the clock and free to evolve, and the clock and morphogenesis of the PSM exhibit developmental modularity. The precise topology of this space remains unclear and poses an intriguing question for further study. We suggest that this modularity may be responsible for the broad diversity of segment number that we observe in the vertebrates (*Richardson et al., 1998*) and indeed is a present source of evolvability in the clade. We note that the evolution of modularity is an active area of research (*Wagner and Altenberg, 1996*; *Wagner et al., 2007*), and while several well-characterised examples of modularity exist (*Monteiro et al., 2003*; *Mallarino et al., 2011*; *Verd et al., 2019*), there are few examples to our knowledge where the existence of such modularity relies on a handful of experimentally tractable parameters. We therefore suggest that studying the evolution of delta-notch signalling in vertebrates, and the evolutionary origin of the PSM and somitogenesis, may be an illuminating model paradigm for this field.

## Methods
### Computational model of the segmentation clock and PSM
To simulate the movements of cells within the PSM and the dynamics of the segmentation clock, we used the model of *Uriu et al., 2021*, who sought to explain defective somite patterning after loss of Notch-Delta signalling. Here, an abstract phase-oscillator Kuramoto model is used to describe intracellular clock oscillations and cell-cell phase coupling. Cells are confined within a three-dimensional tissue domain whose geometry approximates that of the PSM.

This model is suitable for our purposes because it provides a three-dimensional model of cell movements within the PSM, allowing us to alter morphogenesis in order to test hypotheses. Additionally, as the segmentation clock here is described as an abstract phase-oscillator, the model has broad applicability to different vertebrate species which are known to differ in the structure and regulation of their segmentation clock gene regulatory networks (*Krol et al., 2011*). Finally, the validity of the model has been shown in several ways via comparison with data for zebrafish (*Webb et al., 2016*; *Liao et al., 2016*; *Uriu et al., 2021*) and uses parameters experimentally inferred from zebrafish embryos (*Riedel-Kruse et al., 2007*; *Soroldoni et al., 2014*; *Uriu et al., 2021*). This gives an accurate read-out of the effect of changing morphogenesis on the segmentation clock of a vertebrate.

The model is described elsewhere (*Uriu et al., 2021*), but for completeness we include an account of it here. For clarity, vector variables are shown in bold, and scalar variables in normal font.

### Tissue geometry and frame of reference
The model assumes that the posterior tip of the PSM is held in an inertial frame of reference and models the posterior-ward elongation of the PSM by movement of cells towards the tissue anterior.

Cells are confined within a horseshoe-shaped domain resembling the PSM. This domain is comprised of two cylindrical subdomains and one half-toroid subdomain, each representing the lateral sides of the PSM and tailbud, respectively. A schematic can be seen in *Figure 9*. Cells are free to move between each subdomain and leave the tissue at the anterior at $x = x_a$.

The centre of the toroid subdomain is denoted by the coordinate vector $(X_c, Y_c, Z_c)^T$, and the anterior of the PSM is demarcated by the $x$-coordinate $x = x_a$ (see *Figure 9*). The major radius of the torus is denoted by $R$ and the minor radius (and the radius of the two cylindrical subdomains) is denoted by $r_T$ (see *Figure 9*). The resulting length of the PSM in the $x$ direction, $L$, is therefore given by $L = r_T + R + X_c - x_a$.

### Cell movement
To approximate cell mixing and the advection of cells out of the PSM towards the anterior (in the chosen tailbud-inertial frame of reference), cell movement in the model is described by an equation of

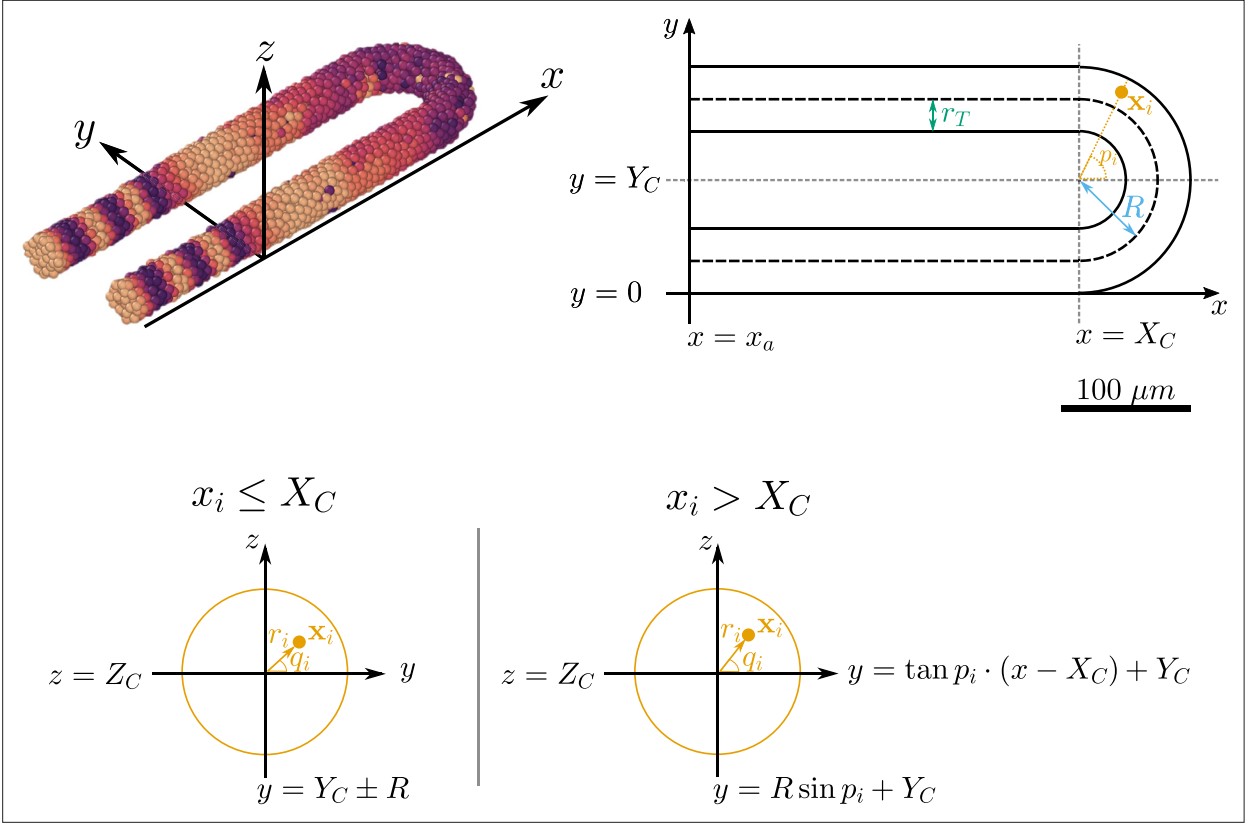

**Figure 9.** Geometry of the pre-somitic mesoderm (PSM) assumed in the present model. Top left: Major axes $(x, y, z)$ used in the model. $x$ corresponds to the anterior-posterior axis of the embryo and increases towards the tissue posterior, $y$ corresponds to the left-right axis and increases to the right-hand side of the tissue, and $z$ corresponds to the dorsal-ventral axis of the embryo. Top right: Schematic of the PSM in the $xy$ plane. The PSM is comprised of two cylinders, centred at $y = r_T$ and $y = 2R + r_T$, respectively, with radius $r_T$. The 'tailbud' is represented as a half-torus subdomain centred at $\mathbf{x} = (X_C, Y_c, Z_c)^T$, with minor radius $r_T$ and major radius $R$. Bottom: Cross sections of the tissue showing how a point $\mathbf{x}_i$ is assigned the polar coordinates $r_i$ and $q_i$, in both the PSM cylinders ($x_i \leq X_c$) and the half-toroid tailbud ($x_i > X_c$). Adapted from *Uriu et al., 2021*.

motion that incorporates cell advection, cell volume exclusion, random cell motion, and a boundary force confining cells within the tissue domain (*Equation 1*).

Specifically, each cell is assigned a position vector $\mathbf{x}_i \in \mathbb{R}^3$, where $\mathbf{x}_i = (x_i, y_i, z_i)^T$. This position $\mathbf{x}_i$ corresponds to the cell centre. Cells are assumed to be spheres with diameter $d_c$. The equation of motion for cell $i$ is

$$\frac{d\mathbf{x}_i}{dt} = \mathbf{v}_d(x_i) + v_0(x_i)\mathbf{n}_i(t) + \sum_{j=1, j \neq i}^{N} \mathbf{F}(\mathbf{x}_i, \mathbf{x}_j) + \mathbf{F}_b(\mathbf{x}_i), \tag{1}$$

where $\mathbf{v}_d(x_i)$ represents the advection of the cell towards the PSM anterior, $v_0(x_i)\mathbf{n}_i(t)$ the intrinsic random motion of the cell, $\mathbf{F}(\mathbf{x}_i, \mathbf{x}_j)$ the repulsion force between cell $i$ and cell $j$, and $\mathbf{F}_b(\mathbf{x}_i)$ the boundary force confining the cell within the tissue.

The advection velocity $\mathbf{v}_d(x_i)$ is given by

$$\mathbf{v}_d(x_i) = (-v_d(x_i), 0, 0)^T, \tag{2}$$

where the function $v_d(x_i)$ describes the local strain rate along the AP axis:

$$v_d(x_i) = \begin{cases} -\dfrac{v_a - v_p(1 - x_q)}{x_q} \cdot \dfrac{x_i - x_a}{L} + v_a & \dfrac{x_i - x_a}{L} \leq x_q, \\[4mm] -\dfrac{v_p(x_i - x_a)}{L} + v_p & \dfrac{x_i - x_a}{L} > x_q, \end{cases} \tag{3}$$

The model assumes that the random cell motility observed within the PSM is intrinsic, i.e., cells would exhibit random motility in isolation from the tissue. This is modelled using the direction vector $\mathbf{n}_i(t)$, which evolves according to a random walk on the surface of the unit sphere specified by the following differential equation:

$$\frac{d\mathbf{n}_i}{dt} = \sqrt{2D}\xi_{\phi_i}(t)\mathbf{m}_x + \sqrt{2D}\xi_{\varphi_i}(t)\mathbf{m}_y - 2D\mathbf{n}_i(t), \tag{4}$$

where the orthogonal vectors (for $\mathbf{e}_z = (0, 0, 1)^T$)

$$\mathbf{m}_x = \frac{\mathbf{n}_i(t) \times \mathbf{e}_z}{|\mathbf{n}_i(t) \times \mathbf{e}_z|}, \quad \mathbf{m}_y = \frac{\mathbf{m}_x \times \mathbf{n}_i(t)}{|\mathbf{m}_x \times \mathbf{n}_i(t)|}, \tag{5}$$

together define a plane tangent to the unit sphere, and $\xi_{\phi_i}(t)$, $\xi_{\varphi_i}(t)$ are white Gaussian noise terms that satisfy, for any $t$, $t'$, $\langle\xi_{\phi_i}(t)\rangle = \langle\xi_{\varphi_i}(t)\rangle = \langle\xi_{\phi_i}(t)\xi_{\varphi_j}(t)\rangle = 0$, and $\langle\xi_{\phi_i}(t)\xi_{\phi_j}(t)\rangle = \langle\xi_{\varphi_i}(t)\xi_{\varphi_j}(t)\rangle = \delta_{ij}\delta(t - t')$. A derivation for *Equation 4* is given by *Uriu et al., 2021*.

To model the increase in random cell motion towards the PSM posterior, the magnitude of intrinsic random cell movement, $v_0(x)$, increases towards the posterior according to *Equation 6*:

$$v_0(x_i) = \frac{v_s}{1 + \left(\dfrac{1 - \dfrac{x_i - x_a}{L}}{X_v}\right)^h}, \tag{6}$$

The parameters $v_s$, $X_v$, and $h$ determine the maximum magnitude of intrinsic cell motion, profile inflexion point, and steepness of the profile, respectively. By varying these parameters, one can create a variety of motility profiles with different shapes and amplitudes (see *Figure 3B*). Here, the *x*-position of cell $i$ is normalised according to the length of the PSM $L$, and so the profile of cell motility scales with the length of the PSM. This is a valid assumption as the length of the PSM is thought to be partially specified by FGF signalling (*Simsek and Özbudak, 2018*), and the random motility of PSM cells is also under the control of FGF (*Bénazéraf et al., 2010*).

Cell volume exclusion is encoded by repulsion of two cells if their centres are within a cell diameter $d_c$ of one another, according to the cell-cell repulsion force $\mathbf{F}(\mathbf{x}_j - \mathbf{x}_i)$:

$$\mathbf{F}(\mathbf{x}_j - \mathbf{x}_i) = F(\mathbf{x}_j - \mathbf{x}_i)\frac{\mathbf{x}_j - \mathbf{x}_i}{|\mathbf{x}_j - \mathbf{x}_i|}, \tag{7}$$

where the magnitude of the cell-cell repulsion force $F(\mathbf{x}_j - \mathbf{x}_i)$ is given by

$$F(\mathbf{x}_j - \mathbf{x}_i) = \begin{cases} \mu\left(\dfrac{|\mathbf{x}_j - \mathbf{x}_i|}{d_c} - 1\right) & |\mathbf{x}_j - \mathbf{x}_i| \leq d_c, \\ 0, & |\mathbf{x}_j - \mathbf{x}_i| > d_c, \end{cases} \tag{8}$$

and the parameter $\mu$ controls the strength of cell-cell repulsion throughout the tissue. For computational simplicity, cell-cell adhesion is not encoded within this model.

The form of the boundary force $\mathbf{F}_b(\mathbf{x}_i)$ varies depending on whether the cell $i$ is in one of the lateral PSM cylindrical subdomains or in the half-toroid tailbud (*Figure 9*). If cell $i$ lies within the cylinders, one can use the cylindrical coordinates for cell $i$, $r_i$, and $q_i$ (see *Figure 9*), to naturally define the boundary force:

$$\mathbf{F}_b(\mathbf{x}_i) = \begin{pmatrix} 0 \\ -\mu_b e^{-\frac{\delta y}{r_b}}\cos q_i \\ -\mu_b e^{-\frac{\delta z}{r_b}}\sin q_i \end{pmatrix}, \tag{9}$$

where

$$\delta y = |(r_T - r_i)\cos q_i|, \tag{10}$$

$$\delta z = |(r_T - r_i)\sin q_i|. \tag{11}$$

The magnitude of the boundary force is denoted by $\mu_b$ and the parameter $r_b$ determines the length-scale of the boundary force. In the half-toroid tailbud the position $\mathbf{x}_i$ of each cell $i$ can be described according to the coordinates $r_i$, $p_i$, and $q_i$:

$$\mathbf{x}_i = \begin{pmatrix} X_c + R\cos p_i + r_i \cos p_i \cos q_i \\ Y_c + R\sin p_i + r_i \sin p_i \cos q_i \\ Z_c + r_i \sin q_i \end{pmatrix}. \tag{12}$$

One can then naturally define

$$\delta x = |(r_T - r_i)\cos p_i \cos q_i|, \tag{13}$$
$$\delta y = |(r_T - r_i)\sin p_i \cos q_i|, \tag{14}$$
$$\delta z = |(r_T - r_i)\sin q_i|, \tag{15}$$

describing the distance of cell $i$ from the tissue surface in the $x$, $y$, and $z$ directions, respectively. The boundary force then becomes

$$\mathbf{F}_b(\mathbf{x}_i) = \begin{pmatrix} -\mu_b e^{-\frac{\delta x}{r_b}}\cos p_i \cos q_i \\ -\mu_b e^{-\frac{\delta y}{r_b}}\sin p_i \cos q_i \\ -\mu_b e^{-\frac{\delta z}{r_b}}\sin q_i \end{pmatrix}. \tag{16}$$

Note that in both definitions of $\mathbf{F}_b(\mathbf{x}_i)$, the tissue subdomains are open-ended, i.e., cells are free to move between subdomains and out of the PSM at the anterior.

For further details we direct the reader to consult *Uriu et al., 2021*.

## Cell addition

As cells are removed from the tissue by advection, new cells must be added to replenish the PSM. Loss of cells is monitored by comparing the total density of cells in each of the subdomains with a target density, $\rho_0$. If the density in any of these subdomains falls below $\rho_0$, a new cell is added to that subdomain. If the density of more than one of the subdomains falls below $\rho_0$ then a cell is only added to the subdomain with the lowest density. The position and phase of the nascent cell both vary across different simulations discussed here. Below is a description of how these quantities vary.

In simulations where cell addition is 'Random', cells are added with a random position within each subdomain, with a random phase, i.e., if a cell is added within one of the cylindrical subdomains, it is given a random $x_i \in [x_a + x_d, X_c]$, a random $r_i \in [0, r_T]$, and $q_i \in [0, 2\pi)$. Alternatively, if the cell is added within the half-toroid subdomain, it is given a random $p_i \in [-\pi/2, \pi/2]$, and random $r_i$ and $q_i$, defined as above. The cell is also assigned a random phase $\theta_i \in [0, 2\pi)$. This method of cell addition is unbiased in terms of its effect on the model but is a biologically implausible scenario of cell ingression, so we use it here as a negative control.

In simulations where we seek to more accurately model cell ingression, cells are added on the cell surface, i.e., $r_i = r$. Additionally, the surface on which cells can be added is restricted. For the half-toroid subdomain, we restrict cell addition to the dorsal quarter of the subdomain, i.e., $q_i \in [\pi/4, 3\pi/4]$, $p_i \in [-\pi/2, \pi/2]$. For the cylindrical subdomains, we restrict addition to the bottom quarter of the surface, i.e., $q_i \in [5\pi/4, 7\pi/4]$. To more accurately model conditions of cell ingression, combinations of these three modes of cell addition are employed (*Figure 2*).

In newly added cells the direction vector $\mathbf{n}$ is initialised with a random vector of unit length. For simulations where cell division is present, cells are added with a random cell cycle phase $\tau \in [0, T_G + T_M)$.

## Coupled oscillator model of the segmentation clock

Here, only the phase of each cell's segmentation clock is considered, and oscillation amplitudes are neglected. This simplification is appropriate when oscillator coupling is weak (*Kuramoto, 1984*). Here,

oscillators have a spatially dependent intrinsic frequency $\omega(x_i)$ and couple to adjacent cells (defined as those within $d_c$, a cell's diameter) with a magnitude controlled by the scalar value $\kappa$:

$$\frac{d\theta_i}{dt} = \omega(x_i) + \frac{\kappa}{N_i(t)} \sum_{|\mathbf{x}_j - \mathbf{x}_i| \leq d_c} \sin(\theta_j - \theta_i) + \sqrt{2D_\theta}\xi_{\theta_i}(t). \tag{17}$$

Oscillations are noisy, with phase noise intensity $D_\theta$. The term $\xi_{\theta_i}(t)$ describes Gaussian noise as before. $N_i(t)$ is the number of cells within a radius of length $d_c$ of cell $i$ at time $t$, i.e., $N_i(t) = |\{j \mid |\mathbf{x}_j(t) - \mathbf{x}_i(t)| \leq d_c\}|$.

The intrinsic frequency $\omega(x_i)$ decreases towards the PSM anterior, creating a frequency gradient and travelling phase waves that move towards the anterior (*Soroldoni et al., 2014*). The intrinsic frequency is defined as:

$$\omega(x_i) = \omega_0\left(\sigma + (1 - \sigma) \cdot \frac{1 - e^{-\frac{k(x - x_a)}{L}}}{1 - e^{-k}}\right), \tag{18}$$

where $\omega_0 > 0$ corresponds to the frequency of cells at the posterior of the tissue, $\sigma > 0$ the fold-change in frequency at the anterior compared to the posterior, and $k > 0$ controls the shape of the gradient. For simplicity we assume that $\omega(x_i)$ is constant over time, as cells at the PSM posterior are thought to maintain a constant frequency over time (*Giudicelli et al., 2007*; *Webb et al., 2016*).

To simulate the effect of the wavefront, where the clock phase is interpreted by cells to pre-pattern somite boundaries (*Cooke and Zeeman, 1976*), beyond $x = x_a$ oscillations arrest, forming stable patterns of phase (*Figure 2A*), i.e., $d\theta_i/dt = 0$ if $x_i < x_a$.

## Modelling the segmentation clock with a phase coupling delay

For simulations where we seek to model the effects of a coupling delay between cells, the phase of cells evolves according to

$$\frac{d\theta_i}{dt} = \omega(x_i) + \frac{\kappa}{N_i(t)} \sum_{|\mathbf{x}_j - \mathbf{x}_i| \leq d_c} \sin(\theta_j(t - \tau) - \theta_i(t)) + \sqrt{2D_\theta}\xi_{\theta_i}(t), \tag{19}$$

where $\tau$ is a constant representing the temporal delay in exchange of phase information between cells, after *Morelli et al., 2009*. In such simulations we initialise the phase of each newly added cell with the initial conditions $\theta_i(t \leq t_{add}) = 0$, where $t_{add}$ is the time at which cell $i$ was added to the tissue.

## Cell division and segmentation clock arrest

During M-phase of the cell cycle, the segmentation clock arrests (*Horikawa et al., 2006*; *Delaune et al., 2012*). To simulate this, we assign each cell $i$ a cell cycle phase $\tau \in [0, T_G + T_M)$, where $T_G$ and $T_M$ are the durations of G1-G2 and M phases in minutes, respectively, and evolve $\tau$ according to

$$\tau = (t - t_{addition}) \mod (T_G + T_M), \tag{20}$$

where $t_{addition}$ denotes the time at which the cell was added.

To simulate the arrest of segmentation clock gene expression during M-phase of the clock, we evolve $\theta_i$ according to

$$\frac{d\theta_i}{dt} = H(T_G - \tau_i) \cdot \left(\omega(x_i) + \frac{\kappa}{N_i(t)} \sum_{|\mathbf{x}_j - \mathbf{x}_i| \leq d_c} \sin(\theta_j - \theta_i) + \sqrt{2D_\theta}\xi_{\theta_i}(t)\right), \tag{21}$$

where $H(x)$ is the Heaviside step function:

$$H(x) = \begin{cases} 1 & x > 0, \\ 0 & x \leq 0. \end{cases} \tag{22}$$

For simulations where we forbid cells from coupling their phase to neighbours undergoing mitosis, the evolution of $\theta_i$ is given by

$$\frac{d\theta_i}{dt} = H(T_G - \tau_i) \cdot \left( \omega(x_i) + \frac{\kappa}{N_i(t)} \sum_{|\mathbf{x}_j - \mathbf{x}_i| \leq d_c, \ \tau_j < T_G} \sin(\theta_j - \theta_i) + \sqrt{2D_\theta}\xi_{\theta_i}(t) \right). \tag{23}$$

If $\tau_i \geq T_G + T_M$ then we consider the cell cycle complete and a new cell is added at a random position of length $d_{new}$ away from cell $i$, i.e.,

$$\mathbf{x}_{N_{total}+1} = \mathbf{x}_i + d_{new} \begin{pmatrix} \cos\varphi\sin\phi \\ \sin\varphi\sin\phi \\ \cos\phi \end{pmatrix},$$

for uniformly distributed random $\phi \in [0, \pi]$, $\varphi \in [0, 2\pi)$, where $N_{total}$ is the total number of cells within the tissue. Other variables are inherited from cell $i$, i.e., $n_{N_{total}+1} = n_i$ and $\theta_{N_{total}+1} = \theta_i$. The choice of $d_{new}$ is somewhat arbitrary, however, as this distance decreases the magnitude of the cell-cell repulsion force (*Equation 8*) grows very large, and one must choose a $d_{new}$ sufficiently small to be biologically accurate while large enough to avoid implausible cell velocities due to a very large cell-cell repulsion force. We find that $d_{new} = d_c/10$ is a suitable compromise between these two scenarios.

We are not aware of any direct measurements of $T_G$ within the zebrafish PSM, and so have calculated a value for this parameter based on some assumptions. M-phase within the PSM lasts approximately 15 min, and approximately 8% of cells in the PSM at a single timepoint are in M-phase (*Horikawa et al., 2006*; *Kanki and Ho, 1997*). Cells in M-phase appear to be distributed homogenously in space (*Kanki and Ho, 1997*), and, therefore, cell cycles in the PSM may be independent, in which case the total duration $T_G + T_M$ is approximately equal to 15/0.08 = 187.5 min and $T_G$ = 172.5 min. Unless otherwise stated, we fix $T_G + T_M$ = 187.5 min.

## Modelling compaction-extension

To model the compaction-extension of the PSM (*Thomson et al., 2021*), we shrink the PSM radius $r_T$, and length $L$, over time, concurrent with an increase in density $\rho_0$, and a stepwise decrease in cell diameter $d_c$ (see *Figure 5*). This is derived from the observations of *Thomson et al., 2021*, who observed changes in the PSM height, width, length, density, and cell diameter, during the latter stages of zebrafish somitogenesis. We sought to model this process using the rates of change in these parameters described by *Thomson et al., 2021*, however, as their published rates are in units of change per somite stage (ss), and the rate of somitogenesis is nonlinear in zebrafish (*Schröter et al., 2008*), we needed an interpolation of somite number over time in order to describe the according change in rate over time.

To this, we extracted the data for somite number over time from *Schröter et al., 2008*, using the web app WebPlotDigitizer (*Rohatgi, 2022*) and fitted a function of the form

$$s(t) = 6 + \frac{t}{24.7} - ae^{b(t-300)}, \tag{24}$$

for unknown parameters $a$, $b$, using the Julia package LsqFit.jl. We obtained values of $a = 0.5001$ and $b = 0.0049$ using this process. Using this interpolation of somite number over time, we then could change the tissue radius, length, cell diameter, and density, using the values published in *Thomson et al., 2021*. As this only occurs in the latter stages of somitogenesis (measured only from 16ss onwards), we hold these variables constant before the time at which 16 somites are formed in zebrafish.

The equations describing change in tissue length ($L$), wavefront position ($x_a$), and anterior limit of cell addition therefore become:

$$L(t) = \begin{cases} L_0 & s(t) < 16, \\ L_0 - m_a(s(t) - 16) & s(t) \geq 16, \end{cases} \tag{25}$$

$$x_a(t) = \begin{cases} x_{a0} & s(t) < 16, \\ x_{a0} + m_a(s(t) - 16) & s(t) \geq 16, \end{cases} \tag{26}$$

$$x_d(t) = x_a(t) + 100, \tag{27}$$

where $m_a > 0$ denotes the rate at which the PSM shrinks in length, and $L_0 > 0$ and $x_{a0} > 0$ the initial values for $L$ and $x_a$ before shrinking, respectively. The equation describing the change in radius $r_T$ is

$$r_T(t) = \begin{cases} r_0 & s(t) < 16, \\ r_0 - m_r(s(t) - 16) & s(t) \geq 16, \end{cases} \tag{28}$$

where $m_r > 0$ denotes the rate at which the PSM shrinks in radius, and $r_0 > 0$ the initial value for $r_T$ before shrinking. In their work *Thomson et al., 2021*, reported that the PSM shrinks in height more rapidly at the PSM posterior than at the anterior, however, the rate of shrinkage of the PSM in width (along the medio-lateral axis) was the same across the anterior and posterior halves of the PSM (*Thomson et al., 2021*). As our model assumes the tissue is as wide as it is tall, we take only the rate of shrinkage in the medio-lateral axis in our value for $m_r$, and calculate $m_r$ by taking the mean of the two (similar) rates published by *Thomson et al., 2021*, for the anterior and posterior halves of the PSM.

The functions describing change in tissue density $\rho$ and cell diameter $d_c$ are similarly

$$\rho(t) = \begin{cases} \rho_{t0} & s(t) < 16, \\ \rho_{t0} + m_\rho(s(t) - 16) & s(t) \geq 16, \end{cases} \tag{29}$$

and

$$d_c(t) = \begin{cases} d_{c0} & s(t) < 16, \\ d_{c0} - m_{d_c}(s(t) - 16) & s(t) \geq 16, \end{cases} \tag{30}$$

where $m_\rho > 0$ and $m_{d_c} > 0$ represent the rates of change in density and cell diameter, respectively, and $\rho_{t0} > 0$ and $d_{c0} > 0$ the initial values for density and cell diameter, respectively, before shrinking. In their work *Thomson et al., 2021*, only report the cell diameter for 16ss and 26ss zebrafish embryos, but we choose to shrink the diameter of cells at the rate they describe, measured between these two timepoints. Our results are insensitive to choice of a function for $d_c(t)$ where cells continue shrinking in diameter at the same rate past 26ss, or not (see *Figure 5—figure supplement 1*).

## Initial conditions

To initialise the simulation, $N$ cells are generated with random positions

$$\mathbf{x}_i = \begin{pmatrix} x_i \\ r_T + r_i \cos q_i \\ Z_c + r_i \sin q_i \end{pmatrix},$$

and a further $N$ with positions

$$\mathbf{x}_i = \begin{pmatrix} x_i \\ r_T + 2R + r_i \cos q_i \\ Z_c + r_i \sin q_i \end{pmatrix},$$

and then a further $N_{TB}$ with positions

$$\mathbf{x}_i = \begin{pmatrix} X_c + R \cos p_i + r_i \cos p_i \cos q_i \\ Y_c + R \sin p_i + r_i \sin p_i \cos q_i \\ Z_c + r_i \sin q_i \end{pmatrix},$$

**Table 1.** Parameter values used for simulations, unless otherwise stated.

| Symbol | Value | Units | Source(s) | Symbol | Value | Units | Source(s) |
|---|---|---|---|---|---|---|---|
| $X_c$ | 300 | μm | Uriu et al., 2021 | $L$ | $X_c - x_a + R + r_T$ | μm | Uriu et al., 2021 |
| $Y_c$ | 85 | μm | Uriu et al., 2021 | $x_a$ | 0 | μm | Uriu et al., 2021 |
| $Z_c$ | 25 | μm | Uriu et al., 2021 | $x_d$ | 100 | μm | Uriu et al., 2021 |
| $R$ | 60 | μm | Uriu et al., 2021 | $x_q$ | 0.7 | – | Uriu et al., 2021 |
| $r_T$ | 25 | μm | Uriu et al., 2021 | $d_c$ | 11 | μm | Uriu et al., 2021 |
| $v_a$ | 1.67 | μm·min⁻¹ | Uriu et al., 2021 | $D$ | 0.026 | min⁻¹ | Uriu et al., 2021 |
| $v_p$ | 3 | μm·min⁻¹ | Uriu et al., 2021 | μ | 8.71 | μm·min⁻¹ | Uriu et al., 2021 |
| $X_v$ | 0.4 | – | Uriu et al., 2021 | $\mu_b$ | 20 | μm·min⁻¹ | Uriu et al., 2021 |
| $h$ | 3 | – | Uriu et al., 2021 | $r_b$ | 1 | μm | Uriu et al., 2021 |
| $v_s$ | 1 | μm·min⁻¹ | Uriu et al., 2021 | $\rho_0$ | 0.0015 | μm⁻³ | Uriu et al., 2021 |
| $L_0$ | 325 | μm | Thomson et al., 2021 | $T_G$ | 187.5–15 | min | Kanki and Ho, 1997, Horikawa et al., 2006 |
| $m_a$ | 14.1 | μm ·somite⁻¹ | Thomson et al., 2021 | $T_M$ | 15 | min | Horikawa et al., 2006 |
| $r_0$ | 27.6 | μm | Thomson et al., 2021 | κ | 0.07 | min⁻¹ | Riedel-Kruse et al., 2007 |
| $m_r$ | 1.1625 | μm ·somite⁻¹ | Thomson et al., 2021 | $\omega_0$ | 0.2094 | min⁻¹ | Soroldoni et al., 2014, Uriu et al., 2021 |
| $d_{c0}$ | 9.2 | μm | Thomson et al., 2021 | σ | 0.66 | – | Uriu et al., 2021 |
| $m_{d_{c0}}$ | 0.2 | μm ·somite⁻¹ | Thomson et al., 2021 | $k$ | 3.07 | – | Uriu et al., 2021 |
| $\rho_{t0}$ | 0.002123 | μm⁻³ | Thomson et al., 2021 | $D\theta$ | 0.0013 | min⁻¹ | Riedel-Kruse et al., 2007 |
| $m_\rho$ | 0.000121 | μm ·somite⁻¹ | Thomson et al., 2021 | τ | 21 | min | Herrgen et al., 2010 |

for $x_i \in [x_a, X_c]$, $p_i \in [-\pi/2, \pi/2]$, $q_i \in [0, 2\pi]$, and $r_i \in [0, r_T]$, where $N = \lfloor \rho_0 \pi r_T^2 X_c \rfloor$ and $N_{TB} = \lfloor \rho_0 \pi^2 r_T^2 R \rfloor$. As this generates more cells near the tissue mid-line than at the periphery, in order to homogenise local density the tissue is 'relaxed' for 10 min, whereby $\mathbf{x}_i$ evolves according to

$$\frac{d\mathbf{x}_i}{dt} = v_0(x_i)\mathbf{n}_i(t) + \sum_{j=1, j\neq i}^{N} \mathbf{F}(\mathbf{x}_i, \mathbf{x}_j) + \mathbf{F}_b(\mathbf{x}_i), \tag{31}$$

and $\mathbf{n}_i(t)$ evolves according to *Equation 4*. This evenly distributes cells throughout the tissue.

When cells are initialised prior to relaxation, $n_i(t)$ is assigned for each cell at random, i.e., $\mathbf{n}_i(0) = (\cos\varphi_i \sin\phi_i, \sin\varphi_i \sin\phi_i, \cos\phi_i)^T$ for random $\varphi_i \in [0, 2\pi)$, $\phi_i \in [0, \pi]$. The segmentation clock phase $\theta_i$ is also assigned as a constant value $\theta_i(0) = 3\pi/2$. For simulations with a coupling delay, we initialise clock phase with $\theta_i(t \leq 0) = 3\pi/2$.

## Parameter values
Parameter values, and their source, are shown in *Table 1*.

## Implementation
Equations for cell position (*Equation 1*) and phase (*Equation 23*) are solved from the initial conditions outlined above across a 1000 min time span using an forward Euler scheme in Julia v1.8.2, using a

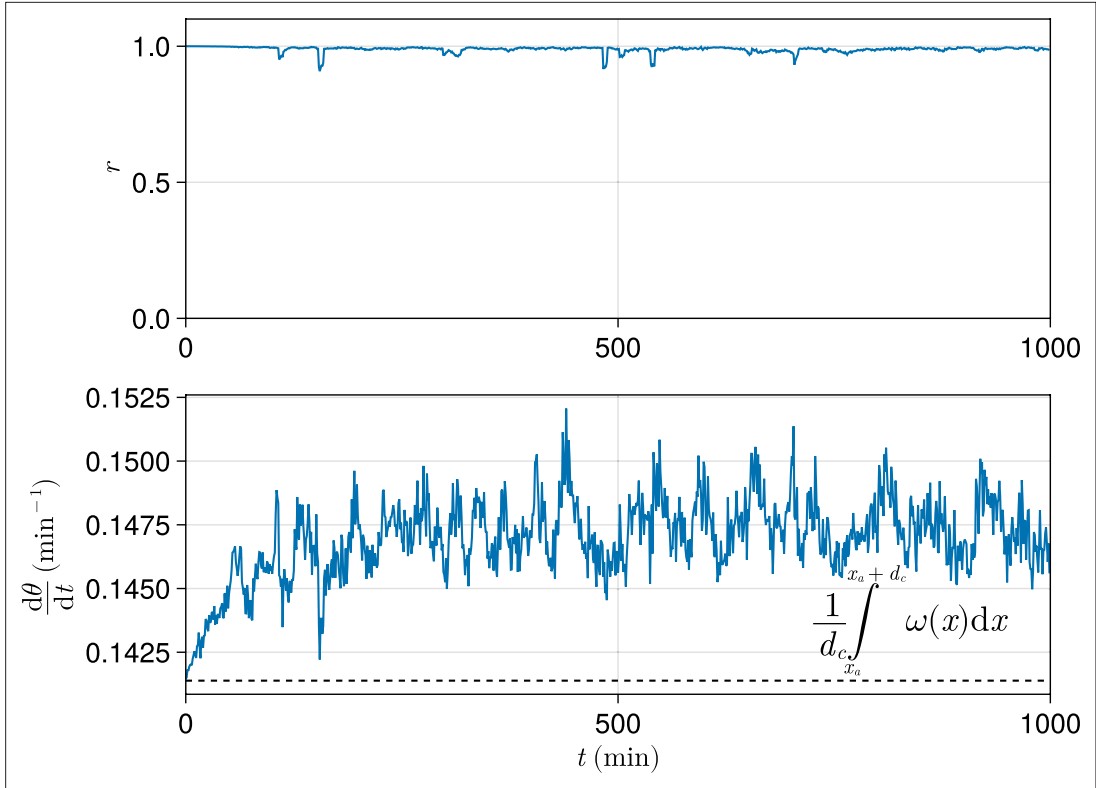

**Figure 10.** Exemplar dynamics of synchrony and mean anterior frequency. Top: Trace of synchrony $r$ for a $d_c$-wide domain of cells at the left-hand anterior edge of the pre-somitic mesoderm (PSM) over 1000 min. Data drawn from a single simulation with random cell addition, using parameters as per *Uriu et al., 2021*. Bottom: Trace of mean frequency $d\theta/dt$ for a $d_c$-wide domain of cells at the left-hand anterior edge of the PSM over 1000 min. Data drawn from the same simulation as above. Plotted with a black dashed line is the average intrinsic frequency $\omega(x)$ across the domain, calculated using the formula shown.

time step $dt$ = 0.01 min. Code for simulation and analysis can be found at https://github.com/jewh/ModularityPSMClock (copy archived at *Hammond, 2024*).

## Analysis and metrics

Here, we measure clock dynamics using two metrics. The first is the synchrony of the segmentation clock across cells, denoted $r$, calculated by

$$re^{i\psi} = \frac{1}{N}\sum_{j=1}^{N} e^{i\theta_j},$$  (32)

which is sometimes referred to as the Kuramoto phase order parameter. When $r \approx 0$, oscillations are asynchronous, and when $r \approx 1$, oscillations are synchronous.

The second metric is the average (mean) frequency of cells in a given domain, where the frequency $d\theta/dt$ is simply the value of *Equation 17*. In cases where we are interested in changes in frequency upon varying model parameters, we find it convenient to define

$$\Delta \frac{d\theta}{dt} = \frac{d\theta}{dt} - \frac{1}{d_c}\int_{x_a}^{x_a+d_c} \omega(x)dx,$$  (33)

in order to highlight changes in frequency.

In order to analyse the output of a simulation in terms of one value, for many analyses we calculate $r$, $d\theta/dt$, or $\Delta d\theta/dt$, for the cells within one cell diameter of the anterior (i.e. $x_a \leq x_i \leq x_a + d_c$). We do this because at the wavefront oscillations arrest and pre-pattern somites, and therefore the frequency and synchrony of oscillations immediately adjacent to the PSM anterior will determine the somite length and precision of pre-patterning. We thus regard the values of $r$, $d\theta/dt$, and $\Delta d\theta/dt$ as proxies

for phenotype exhibited by the simulation. As discussed above, unless stated otherwise we measure clock dynamics after 1000 min, as we find this sufficient to ensure that the dynamics of the clock have reached a steady state that we deem would be reached in vivo (see *Figure 10*).

## Acknowledgements

We thank Koichiro Uriu for sharing his code. We thank Usha Kadiyala for comments regarding estimates for $T_G$. We thank Ben Steventon for feedback on the project. The authors would like to acknowledge the use of the University of Oxford Advanced Research Computing (ARC) facility (http://dx.doi.org/10.5281/zenodo.22558) in carrying out this work. JH was supported by a Natural Environment Research Council DTP studentship (grant NE/S007474/1). This work was supported by a grant from the Simons Foundation (MP-SIP-00001828, REB).

## Additional information

### Funding

| Funder | Grant reference number | Author |
|---|---|---|
| Natural Environment Research Council | Environmental Research DTP grant NE/S007474/1 | James E Hammond |
| John Fell Fund, University of Oxford | 0009780 | Berta Verd |
| European Research Council | 101163722 StG Counts | Berta Verd |
| Simons Foundation | MP-SIP-00001828 | Ruth E Baker |
| Royal Society | RGS\R1\211324 | Berta Verd |

The funders had no role in study design, data collection and interpretation, or the decision to submit the work for publication.

### Author contributions

James E Hammond, Conceptualization, Data curation, Software, Formal analysis, Funding acquisition, Validation, Investigation, Visualization, Methodology, Writing – original draft; Ruth E Baker, Supervision, Funding acquisition, Writing – review and editing; Berta Verd, Conceptualization, Supervision, Funding acquisition, Project administration, Writing – review and editing

### Author ORCIDs

James E Hammond ⓘ https://orcid.org/0000-0001-9603-3410
Ruth E Baker ⓘ https://orcid.org/0000-0002-6304-9333
Berta Verd ⓘ https://orcid.org/0000-0001-9835-009X

Reviewer #1 (Public review): https://doi.org/10.7554/eLife.106316.2.sa1
Reviewer #2 (Public review): https://doi.org/10.7554/eLife.106316.2.sa2
Reviewer #3 (Public review): https://doi.org/10.7554/eLife.106316.2.sa3
Author response https://doi.org/10.7554/eLife.106316.2.sa4

## Additional files

### Supplementary files
MDAR checklist

### Data availability
All code is available at https://github.com/jewh/ModularityPSMClock (copy archived at *Hammond, 2024*).

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
