## [Editor Report · eLife Assessment]

This **valuable** manuscript uses mathematical modeling to address the synchrony of the vertebrate segmentation clock with the developmental processes. The authors use **convincing** arguments to support the idea that this would allow the evolution of flexible body plans and a variable number of segments. This manuscript could be of interest to developmental biologists and systems biologists.

[Editors' note: this paper was reviewed by Review Commons.]

---

## [Referee Report · Reviewer #1 (Public review)]

Summary:

In this manuscript, Hammond et al. study robustness of the vertebrate segmentation clock against morphogenetic processes such as cell ingression, cell movement and cell division to ask whether the segmentation clock and morphogenesis are modular or not. The modularity of these two would be important for evolvability of the segmenting system. The authors adopt a previously proposed 3D model of the presomitic mesoderm (Uriu et al. 2021 eLife) and include new elements; different types of cell ingression, tissue compaction and cell cycles. Based on the results of numerical simulations that synchrony of the segmentation clock is robust, the authors conclude that there is a modularity in the segmentation clock and morphogenetic processes.

The presented results support the conclusion. The manuscript is clearly written.

Major comments from the original round of review:

[Optional] In both the current model and Uriu et al. 2021, coupling delay in phase oscillator model is not considered. Given that several previous studies (e.g. Lewis 2003, Herrgen et al. 2010, Yoshioka-Kobayashi et al. 2020) suggested the presence of coupling delays in Delta-Notch signaling, could the authors analyze the effect of coupling delay on robustness of the segmentation clock against morphogenetic processes?

Significance:

Synchronization of the segmentation clock has been studied by mathematical modeling, but most previous studies considered cells in a static tissue without morphogenesis. In the previous study by Uriu et al. 2021, morphogenetic processes such as cell advection due to tissue elongation, tissue shortening, and cell mobility were considered in synchronization. The current manuscript provides methodological advances in this aspect by newly including cell ingression, tissue compaction and cell cycle. In addition, the authors bring a concept of modularity and evolvability to the field of the vertebrate segmentation clock, which is new. On the other hand, the manuscript confirms that the synchronization of the segmentation clock is robust by careful simulations, but it does not propose or reveal new mechanisms for making it robust or modular. The main targets of the manuscript will be researchers working on somitogenesis and evolutionary biologists who are interested in evolution of developmental systems. The manuscript will also be interested by broader audiences, like developmental biologists, biophysicists, and physicists and computer scientists who are working on dynamical systems.

---

## [Referee Report · Reviewer #2 (Public review)]

Summary:

The manuscript from Hammond et al., investigates the modularity of the segmentation clock and morphogenesis in early vertebrate development, focusing on how these processes might independently evolve to influence the diversity of segment numbers across vertebrates.

Methodology: The study uses a previously published computational model, parameterized for zebrafish, to simulate and analyse the interactions between the segmentation clock and the morphogenesis of the pre-somitic mesoderm (PSM). Their model integrates cell advection, motility, compaction, cell division, and the synchronization of the embryo clock. Three alternative scenarios of PSM morphogenesis were modeled to examine how these changes affect the segmentation clock.

Model System: The computational model system combines a representation of cell movements and the phase oscillator dynamics of the segmentation clock within a three-dimensional horseshoe-shaped domain mimicking the geometry of the vertebrate embryo PSM. The parameters used for the mathematical model are mostly estimated from previously published experimental findings.

Key Findings and Conclusions: (1) The segmentation clock was found to be broadly robust against variations in morphogenetic processes such as cell ingression and motility; (2) Changes in the length of the PSM and the strength of phase coupling within the clock significantly influenced the system's robustness; (3) The authors conclude that the segmentation clock and PSM morphogenesis exhibited developmental modularity (i.e. relative independence), allowing these two phenomena to evolve independently, and therefore possibly contributing to the diverse segment numbers observed in vertebrates.

Major comments from the original round of review:

(1) The key conclusion drawn by the authors (that there is robustness, and therefore modularity, between the morphogenetic cellular processes modeled and the embryo clock synchronization) stems directly from the modeling results appropriately presented and discussed in the manuscript.

The model comprises some strong assumptions, however all have been clearly explained and the parameterization choices are supported by experimental findings, providing biological meaning to the model. Estimated parameters are well explained, and seem reasonable assumptions (from the embryology perspective).

(2) This study, as is, achieves its proposed goal of evaluating the potential robustness of the embryo clock to changes in (some) morphogenetic processes. The authors do not claim that the model used is complete, and they properly identify some limitations, including the lack of cell-cell interactions. Given the recognized importance of cellular physical interactions for successful embryo development, including them in the model would be a significant addition in future studies.

(3) The authors have deposited all the code used for analysis in a public GitHub repository that is updated and available for the research community.

(4) In page 6, the authors justify their choice of clock parameters for cells ingressing the PSM: "As ingressing cells do not appear to express segmentation clock genes (Mara et al. (2007)), the position at which cells ingress into the PSM can create challenges for clock patterning, as only in the 'off' phase of the clock will ingressing cells be in-phase with their neighbors."

However, there are several lines of evidence (in chick and mouse), that some oscillatory clock genes are already being expressed as early as in the gastrulation phase (so prior to PSM ingression) (Feitas et al, 2001 [10.1242/dev.128.24.5139]; Jouve et al, 2002 [10.1242/dev.129.5.1107]; Maia-Fernandes at al, 2024 [10.1371/journal.pone.0297853]).

Question: Is this also true in zebrafish? (I.e. is there any recent experimental evidence that the clock genes are not expressed at ingression, since the paper cited to support this assumption is from 2007).

If they are expressed in zebrafish (as they are in mouse and chick), then the cell addition should have random clock gene periods when they enter the PSM and not start all with a constant initial phase of zero. Probably this will not impact the results since the cells will also be out of phase with their neighbors when they "ingress", however, it will model more closely the biological scenario (and avoid such criticism).

Significance:

GENERAL ASSESSMENT

This study uses a previously published model to simulate alternative scenarios of morphogenetic parameters to infer the potential independence (termed here modularity) between the segmentation clock and a set of morphogenetic processes, arguing that such modularity could allow the evolution of more flexible body plans, therefore partially explaining the variability in the number of segments observed in the vertebrates. This question is fundamental and relevant, yet still poorly researched. This work provides a comprehensive simulation with a model that tries to simplify the many morphogenetic processes described in the literature, reducing it to a few core fundamental processes that allow drawing the conclusions sought. It provides theoretical insight to support a conceptual advance in the field of evolutionary vertebrate embryology.

ADVANCE

This study builds on a model recently published by Uriu et al. (eLife, 2021) that incorporates quantitative experimental data within a modeling framework including cell and tissue-level parameters, allowing the study of multiscale phenomena active during zebrafish embryo segmentation. Uriu's publication reports many relevant and often non-intuitive insights uncovered by the model, most notably the description of phase vortices formed by the synchronizing genetic oscillators interfering with the traveling-wave front pattern.

However, this model can be further explored to ask additional questions beyond those described in the original paper. A good example is the present study, which uses this mathematical framework to investigate the potential independence between two of the modeled processes, thereby extracting extra knowledge from it. Accordingly, the present study represents a step forward in the direction of using relevant theoretical frameworks to quantitatively explore the landscape of complex molecular hypotheses in silico, and with it shed some light on fundamental open questions or inform the design of future experiments in the lab.

The study incorporates a wide range of existing literature on the developmental biology of vertebrates. It comprehensively cites prior work, such as the foundational studies by Cooke and Zeeman on the segmentation clock and the role of FGF signaling in PSM development as discussed by Gomez et al. The literature properly covers the breadth of knowledge in this field.

AUDIENCE

Target audience: This study is relevant for fundamental research in developmental biology, specifically targeting researchers who focus on early embryo development and morphogenesis from both experimental and theoretical perspectives. It is also relevant for evolutionary biologists investigating the genetic factors that influence vertebrate evolution, as well as to computational biologists and bioinformatics researchers studying developmental processes and embryology.

Developmental researchers studying the segmentation clock in other vertebrate model organisms (namely mouse and chick), will find this publication especially valuable since it provides insights that can help them formulate new hypotheses to elucidate the molecular mechanisms of the clock (for example finding a set of evolutionarily divergent genes that might interfere with PSM length).

Additionally, this study provides a set of cellular parameters that have yet to be measured in mouse and chick, therefore guiding the design of future experiments to measure them, allowing the simulation of the same model with sets of parameters from different vertebrate model organisms, therefore testing the robustness of the findings reported for zebrafish.

---

## [Referee Report · Reviewer #3 (Public review)]

Summary:

In this manuscript, Verd and colleagues explored how various biologically relevant factors influence the robustness of clock dynamics synchronization among neighboring cells within the context of somatogenesis, adapting a mathematical model presented by Urio et. al in 2021 in a similar context. Specifically they show that clock dynamics is robust to different biological mechanisms such as cell infusion, cellular motility, compaction-extension and cell-division. On the other hand , the length of Presomitic Mesoderm (PSM) and density of cells in it has a significant role in the robustness of clock dynamics. While the manuscript is well-written and provides clear descriptions of methods and technical details, it tends to be somewhat lengthy.

Major comments from original round of review:

(1) The authors mention that "...the model is three dimensional and so can quantitatively recapture the rates of cell mixing that we observe in the PSM". I am not convinced with this justification of using a 3D model. None of the effects the authors explore in this manuscript requires a three dimensional model or full physical description of the cellular mechanics such as excluded volume interaction etc. A one-dimensional model characterized by cell position along the arclength of PSM and somatic region and segmentation clock phase θ can incorporate all the physics authors described in this manuscript as well as significantly computationally cheap allowing the authors to explore the effect of different parameters in greater detail.

(2) I am not sure about the justification for limiting the quantification of phase synchrony in a very limited (one cell diameter wide) region at one end of the somatic part (Page 33 below Fig. 9). From my understanding of the manuscript, the segments appear in significant length anterior to this region. Wouldn't an ensemble average of multiple such one cell diameter wide regions in the somatic region be a more accurate metric for quantifying synchrony?

(3) While studying the effect of cellular ingression, the authors study three discrete modes-random, DP and DP+LV and show that in the DP+LV mode the clock synchrony becomes affected. I would like the authors to explore this in a continuous fashion from a pure DP ingression to Pure LV ingression and intermediates.

(4) While studying the effect of length and density of cells in PSM on cellular synchrony, the authors restrict to 3 values of density and 6 values of PSM length keeping the other parameter constant. I would be interested to see a phase diagram similar to Fig. 7 in the two dimensional parameter space of L and ρ0. I am curious if a scaling relation exists for the parameter values that partition the parameter space with and without synchrony.

(5) Both in the abstract and introduction, the authors discuss at a great length about the variability in the number of segments. I am curious how the number and width of the segments observed depend on different parameters related to cellular mechanics and the segmentation clock ?

(6) The authors assume that the phase dynamics of the chemical network may be described by an oscillator with constant frequency. For the completeness of the manuscript, the author should discuss in detail, for which chemical networks this is a good assumption.

(7) Figure 3 and the associated text shows no effect of the cellular motility profile in the synchrony of the segmentation clock. This may be moved to the supplementary considering the length of this manuscript.

Significance:

The manuscript answers some important questions in the synchrony of segmentation clock in the vertebrates utilizing a model published earlier. However, the presented result is incomplete in some aspects (points 2 to 5 of section A) and that could be overcome by a more detailed analysis using a simpler one dimensional (point 1 of section A). I believe this manuscript could be of interest to an intersecting audience of developmental biologists, systems biologists, and physicists/engineers interested in dynamical systems.

[Editors' note: the authors have responded comprehensively to the reviews from Review Commons.]

---

## [Author Response]

**Reviewer #1 (Evidence, reproducibility and clarity):**
Summary:In this manuscript, Hammond et al. study robustness of the vertebrate segmentation clock against morphogenetic processes such as cell ingression, cell movement and cell division to ask whether the segmentation clock and morphogenesis are modular or not. The modularity of these two would be important for evolvability of the segmenting system. The authors adopt a previously proposed 3D model of the presomitic mesoderm (Uriu et al. 2021 eLife) and include new elements; different types of cell ingression, tissue compaction and cell cycles. Based on the results of numerical simulations that synchrony of the segmentation clock is robust, the authors conclude that there is a modularity in the segmentation clock and morphogenetic processes. The presented results support the conclusion. The manuscript is clearly written. I have several comments that could help the authors further strengthen their arguments.Major comment:[Optional] In both the current model and Uriu et al. 2021, coupling delay in phase oscillator model is not considered. Given that several previous studies (e.g. Lewis 2003, Herrgen et al. 2010, Yoshioka-Kobayashi et al. 2020) suggested the presence of coupling delays in DeltaNotch signaling, could the authors analyze the effect of coupling delay on robustness of the segmentation clock against morphogenetic processes?

We thank the reviewer for the suggestion. Owing to the computational demands of including such a delay in the model, we cannot feasibly repeat every simulation analysed here in the presence of delay, and would like to note that the increased computational demand that delays put on the simulations is also the reason why Uriu et al 2021 did not include it, as stated in their published exchange with reviewers. However, analogous to our analysis in figure 7, we can analyse how varying the position of progenitor cell ingression affects synchrony in the presence of the coupling delay measured in zebrafish by Herrgen et al. (2010). We show this analysis in a new figure 8 (8B, specifically), on page 21, and discuss its implications in the text on pages 2022. Our analysis reveals that the model cannot recover synchrony using the default parameters used by Uriu et al. (2021) and reveal a much stronger dependence on the rate of cell mixing (vs) than shown in the instantaneous coupling case (cf. figure 7). However, by systematically varying the value of the delay we find that a relatively minor increase in the delay is sufficient to recover synchrony using the parameter set of Uriu et al. (see figure 8C). Repeating this across the three scenarios of cell ingression we see that the combination of coupling strength and delay determine the robustness of synchrony to varying position of cell ingression. This suggests that the combination of these two parameters constrain the evolution of morphogenesis.

Minor comments:- PSM radius and oscillation synchrony are both denoted by the same alphabet r. The authors should use different alphabets for these two to avoid confusion.

We thank the reviewer for spotting this. This has now been changed throughout to rT, as shorthand for ‘radius of tissue’.

- page 5 Figure 1 caption: (x-x_a/L) should be (x-x_a)/L.

We thank the reviewer for spotting this. This has now been corrected.

- Figure 3C: Description of black crosses in the panels is required in the figure legend.

Thank you for spotting this. The legend has now been corrected.

- Figure 3C another comment: In this panel, synchrony r at the anterior PSM is shown. It is true that synchrony at anterior PSM is most relevant for normal segment formation. However, in this case, the mobility profile is changed, so it may be appropriate to show how synchrony at mid and posterior PSM would depend on changes in mobility profile. Is synchrony improved by cell mobility at the region where cell ingression happens?

We thank the reviewer for the suggestion. We have now plotted the synchrony along the AP axis for varying motility profiles, and this can be seen in figure 3 supplement 1, and is briefly discussed in the text on page 11. We show that while the synchrony varies with x-position (as already expected, see figure 2), there is no trend associated with the shape of the motility profile.

- In page 12, the authors state that "the results for the DP and DP+LV cases are exactly equal for L = 185 um, as .... and the two ingression methods are numerically equivalent in the model". I understood that in this case two ingression methods are equivalent, but I do not understand why the results are "exactly" equal, given the presence of stochasticity in the model.

These results can be exactly equal despite the simulations being stochastic because they were both initialised using the same ‘seed’ in the source code. However, we now see that this might be confusing to the reader, and we have re-generated this figure but this time initialising the simulations for each ingression scenario using a different seed value. This is now reflected in the text on page 12 and in figure 4.

- The authors analyze the effect of cell density on oscillation synchrony in Fig. 4 and they mention that higher density increases robustness of the clock by increasing the average number of interacting neighbours. I think it would be helpful to plot the average number of neighbouring cells in simulations as a function of density to quantitatively support the claim.

We thank the reviewer for their suggestion. Distributions of neighbour numbers for exemplar simulations with varying density can now be found in figure 4 supplementary figure 1 and are referred to in the text on page 11.

- The authors analyze the effect of PSM length on synchrony in Fig. 4. I think kymographs of synchrony r as shown in Fig. 2D would also be helpful to show that indeed cells get synchronized while advecting through a longer PSM.

We thank the reviewer for their suggestion and agree that visualising the data in this way is an excellent idea. We have generated the suggested kymographs and added them to figure 4 as supplements 2 and 4, and discussed these results in the text on page 12.

- I understand that cells in M phase can interact with neighboring cells with the same coupling strength kappa in the model, although their clocks are arrested. If so, this aspect should be also mentioned in the main text in page 16, as this coupling can be another noise source for synchrony.

We agree this is an important clarification. We explicitly state this, and briefly justify our choice, in the text on page 16.

- Figure 5-figure supplement 2: panel labels A, B, C are missing.

Thank you for bringing this to our attention. These have now been added.

– Figure 5-figure supplement 3: panel labels A, B, C are missing.

Thank you for bringing this to our attention. These have now been added.

**Reviewer #1 (Significance):**
Synchronization of the segmentation clock has been studied by mathematical modeling, but most previous studies considered cells in a static tissue without morphogenesis. In the previous study by Uriu et al. 2021, morphogenetic processes such as cell advection due to tissue elongation, tissue shortening, and cell mobility were considered in synchronization. The current manuscript provides methodological advances in this aspect by newly including cell ingression, tissue compaction and cell cycle. In addition, the authors bring a concept of modularity and evolvability to the field of the vertebrate segmentation clock, which is new. On the other hand, the manuscript confirms that the synchronization of the segmentation clock is robust by careful simulations, but it does not propose or reveal new mechanisms for making it robust or modular. The main targets of the manuscript will be researchers working on somitogenesis and evolutionary biologists who are interested in evolution of developmental systems. The manuscript will also be interested by broader audiences, like developmental biologists, biophysicists, and physicists and computer scientists who are working on dynamical systems.

We thank the reviewer for their interest in our manuscript and for acknowledging us as one of the first to address the modularity and evolvability of somitogenesis. We hope that this work will encourage others to think about these concepts in this system too.

In the original submission, we identified a high enough coupling strength as the main mechanism underlying the identified modularity in somitogenesis. Since, we have included an analysis of the coupling delay and find that it is the interplay between coupling strength and coupling delay that mediate the identified modularity, allowing PSM morphogenesis and the segmentation clock to evolve independently in regions of parameter space that are constrained and determined by the interplay between these two parameters. We have now added an extra figure (figure 8) where we explore this interplay and have discussed it at length in the last section of the results and in the discussion. We again thank the reviewer for encouraging us to include delays in our analysis.

**Reviewer #2 (Evidence, reproducibility and clarity):**
SUMMARYThe manuscript from Hammond et al., investigates the modularity of the segmentation clock and morphogenesis in early vertebrate development, focusing on how these processes might independently evolve to influence the diversity of segment numbers across vertebrates.Methodology: The study uses a previously published computational model, parameterized for zebrafish, to simulate and analyse the interactions between the segmentation clock and the morphogenesis of the pre-somitic mesoderm (PSM). Their model integrates cell advection, motility, compaction, cell division, and the synchronization of the embryo clock. Three alternative scenarios of PSM morphogenesis were modeled to examine how these changes affect the segmentation clock.Model System: The computational model system combines a representation of cell movements and the phase oscillator dynamics of the segmentation clock within a three-dimensional horseshoe-shaped domain mimicking the geometry of the vertebrate embryo PSM. The parameters used for the mathematical model are mostly estimated from previously published experimental findings.Key Findings and Conclusions: (1) The segmentation clock was found to be broadly robust against variations in morphogenetic processes such as cell ingression and motility; (2) Changes in the length of the PSM and the strength of phase coupling within the clock significantly influenced the system's robustness; (3) The authors conclude that the segmentation clock and PSM morphogenesis exhibited developmental modularity (i.e. relative independence), allowing these two phenomena to evolve independently, and therefore possibly contributing to the diverse segment numbers observed in vertebrates.MAJOR COMMENTS(1) The key conclusion drawn by the authors (that there is robustness, and therefore modularity, between the morphogenetic cellular processes modeled and the embryo clock synchronization) stems directly from the modeling results appropriately presented and discussed in the manuscript. The model comprises some strong assumptions, however all have been clearly explained and the parameterization choices are supported by experimental findings, providing biological meaning to the model. Estimated parameters are well explained and seem reasonable assumptions (from the embryology perspective).

We thank the reviewer for their positive comments about our work

(2) This study, as is, achieves its proposed goal of evaluating the potential robustness of the embryo clock to changes in (some) morphogenetic processes. The authors do not claim that the model used is complete, and they properly identify some limitations, including the lack of cellcell interactions. Given the recognized importance of cellular physical interactions for successful embryo development, including them in the model would be a significant addition in future studies.

We would like to clarify that the model does include cell-cell interactions as cells interact with their neighbours’ clock phase to synchronise and to avoid occupying the same physical space.

(3) The authors have deposited all the code used for analysis in a public GitHub repository that is updated and available for the research community.

We support open source coding practices.

(4) In page 6, the authors justify their choice of clock parameters for cells ingressing the PSM: "As ingressing cells do not appear to express segmentation clock genes (Mara et al. (2007)), the position at which cells ingress into the PSM can create challenges for clock patterning, as only in the 'off' phase of the clock will ingressing cells be in-phase with their neighbours." However, there are several lines of evidence (in chick and mouse), that some oscillatory clock genes are already being expressed as early as in the gastrulation phase (so prior to PSM ingression) (Feitas et al, 2001 [10.1242/dev.128.24.5139]; Jouve et al, 2002 [10.1242/dev.129.5.1107]; Maia-Fernandes at al, 2024 [10.1371/journal.pone.0297853]) Question: Is this also true in zebrafish? (I.e. is there any recent experimental evidence that the clock genes are not expressed at ingression, since the paper cited to support this assumption is from 2007). If they are expressed in zebrafish (as they are in mouse and chick), then the cell addition should have random clock gene periods when they enter the PSM and not start all with a constant initial phase of zero. Probably this will not impact the results since the cells will also be out of phase with their neighbours when they "ingress", however, it will model more closely the biological scenario (and avoid such criticism).

We thank the reviewer for their comments. While it is known that in zebrafish the clock begins oscillating during epiboly and before the onset of segmentation (Riedel-Kruse et al., 2007), to our knowledge no-one has examined whether posteriorly or laterally ingressing progenitor cells express clock genes prior to their ingression into the PSM, which occurs later in development than the first oscillations which give rise to the first somites. We have not found any published evidence of her/hes gene expression in the dorsal donor tissues or lateral tissues surrounding the PSM, however we acknowledge that this has not been actively studied before and our assumption relies on an absence of evidence, rather than evidence of absence.

However, we agree with the reviewer that one should include such an analysis for completeness, and we have now generated additional simulations where progenitor cells ingress with a random clock phase. This data is presented in figure 2 supplement 1 and mentioned in the main text on page 9.

MINOR COMMENTS(1) The citations are appropriate and cover the major labs that have published work related to this study (although with some overrepresentation of the lab that published the model used).

We have cited the vast literature on somitogenesis to the best of our ability and do recognise that the work of the Oates lab appears prominently, but this is probably because their experimental data were originally used to parametrise the model in Uriu et al. 2021.

(2) The text is clear, carefully written, and both the methods and the reasoning behind them are clearly explained and supported by proper citations.

We are very glad to see that the reviewer found that the manuscript was clearly presented.

(3) The figures are comprehensive, properly annotated, with explanatory self-contained legends. I have no comments regarding the presentation of the results.

Thank you

(4) Minor suggestions:a. Page 26: In the Cell addition sub-section of the Methods section, correct all instances where the word domain is used, but subdomain should be used (for clarity and coherence with the description of the model, stated as having a single domain comprising 3 subdomains).

We thank the reviewer for raising this, this is a good point. We have now corrected to ‘subdomain’ where appropriate.

b. Page 32: Table 1. Parameter values used in our work, unless otherwise stated -> Suggestion: Add a column with the individual citations used for each parameter (to facilitate the confirmation of each corresponding reference).

Thank you for the suggstion, we have now done this (see table 1 page 36).

**Reviewer #2 (Significance):**
GENERAL ASSESSMENTThis study uses a previously published model to simulate alternative scenarios of morphogenetic parameters to infer the potential independence (termed here modularity) between the segmentation clock and a set of morphogenetic processes, arguing that such modularity could allow the evolution of more flexible body plans, therefore partially explaining the variability in the number of segments observed in the vertebrates. This question is fundamental and relevant, yet still poorly researched. This work provides a comprehensive simulation with a model that tries to simplify the many morphogenetic processes described in the literature, reducing it to a few core fundamental processes that allow drawing the conclusions seeked. It provides theoretical insight to support a conceptual advance in the field of evolutionary vertebrate embryology.ADVANCEThis study builds on a model recently published by Uriu et al. (eLife, 2021) that incorporates quantitative experimental data within a modeling framework including cell and tissue-level parameters, allowing the study of multiscale phenomena active during zebrafish embryo segmentation. Uriu's publication reports many relevant and often non-intuitive insights uncovered by the model, most notably the description of phase vortices formed by the synchronizing genetic oscillators interfering with the traveling-wave front pattern. However, this model can be further explored to ask additional questions beyond those described in the original paper. A good example is the present study, which uses this mathematical framework to investigate the potential independence between two of the modeled processes, thereby extracting extra knowledge from it. Accordingly, the present study represents a step forward in the direction of using relevant theoretical frameworks to quantitatively explore the landscape of complex molecular hypotheses in silico, and with it shed some light on fundamental open questions or inform the design of future experiments in the lab.The study incorporates a wide range of existing literature on the developmental biology of vertebrates. It comprehensively cites prior work, such as the foundational studies by Cooke and Zeeman on the segmentation clock and the role of FGF signaling in PSM development as discussed by Gomez et al. The literature properly covers the breadth of knowledge in this field.AUDIENCETarget audience | This study is relevant for fundamental research in developmental biology, specifically targeting researchers who focus on early embryo development and morphogenesis from both experimental and theoretical perspectives. It is also relevant for evolutionary biologists investigating the genetic factors that influence vertebrate evolution, as well as to computational biologists and bioinformatics researchers studying developmental processes and embryology.Developmental researchers studying the segmentation clock in other vertebrate model organisms (namely mouse and chick), will find this publication especially valuable since it provides insights that can help them formulate new hypotheses to elucidate the molecular mechanisms of the clock (for example finding a set of evolutionarily divergent genes that might interfere with PSM length). Additionally, this study provides a set of cellular parameters that have yet to be measured in mouse and chick, therefore guiding the design of future experiments to measure them, allowing the simulation of the same model with sets of parameters from different vertebrate model organisms, therefore testing the robustness of the findings reported for zebrafish.
**Reviewer #3 (Evidence, reproducibility and clarity):**
In this manuscript, Verd and colleagues explored how various biologically relevant factors influence the robustness of clock dynamics synchronization among neighboring cells within the context of somatogenesis, adapting a mathematical model presented by Urio et. al in 2021 in a similar context. Specifically they show that clock dynamics is robust to different biological mechanisms such as cell infusion, cellular motility, compaction-extension and cell-division. On the other hand , the length of Presomitic Mesoderm (PSM) and density of cells in it has a significant role in the robustness of clock dynamics. While the manuscript is well-written and provides clear descriptions of methods and technical details, it tends to be somewhat lengthy.Below are the comments I would like the authors to address:(1) The authors mention that "...the model is three dimensional and so can quantitatively recapture the rates of cell mixing that we observe in the PSM". I am not convinced with this justification of using a 3D model. None of the effects the authors explore in this manuscript requires a three dimensional model or full physical description of the cellular mechanics such as excluded volume interaction etc. A one-dimensional model characterized by cell position along the arclength of PSM and somatic region and segmentation clock phase θ can incorporate all the physics authors described in this manuscript as well as significantly computationally cheap allowing the authors to explore the effect of different parameters in greater detail.

One of the main objectives of the work we present in this manuscript is to assess how the evolution of PSM morphogenesis affects, or does not affect, segment patterning. The PSM is a three-dimensional tissue with differing cell rearrangement dynamics along its anterior-posterior axis. In addition, PSM dimension, density, the rearrangement rate, and patterns of cell ingression all vary across vertebrate species, and they are functional, especially cell mixing as it promotes synchronisation and drives elongation. In order to answer questions on the modularity of somitogenesis we therefore consider it absolutely necessary to include a three-dimensional representation of the PSM that captures single cells and their movements. In addition, this will allow us, as Reviewer #2 also pointed out, to reparametrize our model using species-specific data as it becomes available.

While the reviewer is right in that lower dimensional representations would be computationally more efficient, and are generally more tractable, it would not be possible to represent cell mixing in one dimension, as this happens in three dimensions. One could perhaps encode the synchrony-promoting effect of cell mixing via some coupling function κ(x) that increases towards the posterior, however it is unclear what existing biological data one could use to parameterise this function or determine its form. Cell mixing can be modelled in a two-dimensional framework, however this cannot quantitatively recapture the rate of cell mixing observed in vivo, which is an advantage of this model.

Furthermore, it is unclear how one would simulate processes such as compactionextension using a one-dimensional model. The two different scenarios of cell ingression which we consider can also not be replicated in a one-dimensional model, as having a population of cells re-acquiring synchrony on the dorsal surface of the tissue while new material is added to the ventral side, creating asynchrony, is qualitatively different than a one-dimensional scenario where cells are introduced continuously along the spatial axis.

(2) I am not sure about the justification for limiting the quantification of phase synchrony in a very limited (one cell diameter wide) region at one end of the somatic part (Page 33 below Fig. 9). From my understanding of the manuscript, the segments appear in significant length anterior to this region. Wouldn't an ensemble average of multiple such one cell diameter wide regions in the somatic region be a more accurate metric for quantifying synchrony?

Indeed, such a metric (e.g. as that used by Uriu et al. to quantify synchrony along the xaxis) would be more accurate for determining synchrony within the PSM. However, as per the clock and wavefront model of somitogenesis, only synchrony at the very anterior of the PSM (or at the wavefront, equivalently) is functional for somitogenesis and thus evolution. Therefore, we restrict our analysis to the anterior-most region of the PSM. We now further justify this in the main text on page 9.

(3) While studying the effect of cellular ingression, the authors study three discrete modes- random, DP and DP+LV and show that in the DP+LV mode the clock synchrony becomes affected. I would like the authors to explore this in a continuous fashion from a pure DP ingression to Pure LV ingression and intermediates.

We thank the reviewer for this suggestion; this is a very interesting question. We are currently working on a related computational and experimental project to address the question of how PSM morphogenesis can change over evolutionary time to evolve the different modes that we see across species. As part of this work, we are running precisely the simulations suggested by the reviewer to find regions of parameter space in which all the relevant morphogenetic processes can freely evolve. While interesting, this work is however outside the scope of the current manuscript.

(4) While studying the effect of length and density of cells in PSM on cellular synchrony, the authors restrict to 3 values of density and 6 values of PSM length keeping the other parameter constant. I would be interested to see a phase diagram similar to Fig. 7 in the two-dimensional parameter space of L and ρ0. I am curious if a scaling relation exists for the parameter values that partition the parameter space with and without synchrony.

We thank the reviewer for their suggestion and agree that this would constitute an interesting addition to the manuscript. We have now generated these data, which are shown in figure 4 supplement 5 and mentioned on page 13. We see no clear relationship between these two variables when co-varying in the presence of random ingression.

(5) Both in the abstract and introduction, the authors discuss at a great length about the variability in the number of segments. I am curious how the number and width of the segments observed depend on different parameters related to cellular mechanics and the segmentation clock ?

We thank the reviewer for this question. It was not clear to us if this was something the reviewer wants us to address in the study’s background and introduction, or an analysis we should include in the results. Therefore, we have responded to both comprehensively below:

The prevailing conceptual framework for understanding this is the clock and wavefront model (Cooke and Zeeman, 1976), which posits that the somite length is inversely proportional to the frequency of the clock relative to the speed of the wavefront, and that the total number of segments is the relative frequency multiplied by the total duration of somitogenesis.

Experimentally we know that the frequency is determined in part by the coupling strength (Liao, Jorg, and Oates, 2016), and from comparative embryological studies (Gomez et al., 2008; Steventon et al., 2016) we know that changes in the elongation dynamics of the PSM correlate with changes in somite number, presumably by altering the total duration of somitogenesis (Gomez et al., 2009). These changes in elongation are thought to be driven by the changes in cell and tissue mechanics we test in our manuscript.

Within our model, we cannot in general predict how the number of segments responds to changes in either clock parameters or cell mechanical parameters, as we lack understanding of what causes somitogenesis to cease; this is thus not encoded in our model and segmentation can in principle proceed indefinitely. Therefore, we have not performed this analysis.

Similarly, we have not included an analysis of somite length. This is for two reasons: (1) as per the clock and wavefront model, the frequency at the PSM anterior (which we analyse) is equivalent to this measurement, as we assume (in general) the wavefront ($x = x_{a}$) is inertial. (2) the length of the nascent somite is not thought to be of much relevance to the adult phenotype, and by extension evolution. Somites undergo cell division and growth soon after their patterning by the segmentation clock, therefore their final size does not majorly depend on the dynamics of the segmentation clock. Rather, the main function of the clock is to control their number (and polarity).

(6) The authors assume that the phase dynamics of the chemical network may be described by an oscillator with constant frequency. For the completeness of the manuscript, the author should discuss in detail, for which chemical networks this is a good assumption.

We thank the reviewer for their suggestion and now justify this assumption in the methods on page 31.

Such an assumption is appropriate for the segmentation clock, as the clock in the posterior of the PSM is thought to oscillate with a constant frequency, at least for the majority of somitogenesis although the frequency of somite formation slows towards the end of this process in zebrafish (Giudicelli et al., 2007, PLoS Biol.). In addition, PSM cells isolated and cultured in the presence of FGF (thus replicating the signalling environment of the posterior PSM) will continue to exhibit her1 oscillations with an apparently constant frequency (Webb et al., 2016).

We note that such formulations are widely used within the segmentation clock literature (e.g. Riedel-Kruse et al., 2007, Morelli et al., 2009).

(7) Figure 3 and the associated text shows no effect of the cellular motility profile in the synchrony of the segmentation clock. This may be moved to the supplementary considering the length of this manuscript.

Thank you for the suggestion. However, we would argue that the lack of effect is a crucial result when discussing modularity. Reviewer #2 agrees with this assessment.

**Reviewer #3 (Significance):**
The manuscript answers some important questions in the synchrony of segmentation clock in the vertebrates utilizing a model published earlier. However, the presented result is incomplete in some aspects (points 2 to 5 of section A) and that could be overcome by a more detailed analysis using a simpler one dimensional (point 1 of section A). I believe this manuscript could be of interest to an intersecting audience of developmental biologists, systems biologists, and physicists/engineers interested in dynamical systems.